# Clustering of NLRP3 induced by membrane or protein scaffolds promotes inflammasome assembly

Elvira Boršić [1,2], Taja Železnik Ramuta [1], Sara Orehek [1],
Mateja Erdani Kreft [3], Matthias Geyer [4], Roman Jerala [1,5] &
Iva Hafner-Bratkovič [1,3,6] ✉

NLRP3 is a pattern recognition receptor forming an inflammasome in response to diverse pathogen and self-derived triggers, but molecular insights on NLRP3 activation are still lacking. Here, we drive ectopic NLRP3 to different subcellular locations in NLRP3-deficient macrophages to map the spatial activation profile of NLRP3, and find that NLRP3 variants enriched at the organellar membranes respond to canonical triggers similarly to wild-type NLRP3; however, unlike wild-type, these NLRP3 variants can be activated even in the absence of the polybasic phospholipid-binding segment. Mechanistically, membrane or protein scaffolds mediate NLRP3 clustering, which leads to the unfastening of the inactive NACHT domain conformation preceding the activated NLRP3 oligomer formation. Our data thus suggest that scaffold-promoted clustering is an important step in NLRP3 activation, enabling NLRP3 to sense distinct activator-induced cellular anomalies exhibited via lipid or protein assemblies, thereby establishing NLRP3 as the master sensor of perturbations in cell homeostasis.

Inflammasomes are potent drivers of inflammation that protect against invading pathogens but can also aggravate communicable and non-communicable diseases. These multiprotein platforms initiate upon sensor oligomerization, which recruits the adaptor apoptosis-associated speck-like protein containing a CARD (ASC). After polymerization, ASC engages pro-caspase-1 molecules to self-activate. Caspase−1, through the activation of proinflammatory cytokines interleukin (IL)-1β and IL-18, as well as gasdermin D (GSDMD)-driven pyroptosis, induces and perpetuates inflammation that underlies a broad range of diseases from neurodegeneration to rare autoinflammatory genetic syndromes[1–7].

Different sensors can nucleate inflammasomes, but none has attracted as much attention as NLR family pyrin domain-containing protein 3 (NLRP3) due to its involvement in various diseases[8,9]. The activation of NLRP3 is heavily regulated at transcriptional, post-transcriptional, and post-translational levels. Activation of NLRP3 requires two signals, priming, which upregulates NLRP3 and pro-IL-1β expression and enables post-translational modifications of NLRP3[10–13] that facilitate inflammasome assembly upon the action of the inflammasome activator (the second signal). NLRP3 is composed of an N-terminal interaction pyrin domain (PYD), a central NACHT domain (present in NAIP, CIITA, HET-E and TP1), and a C-terminal leucine-rich-repeat (LRR) domain. In the absence of a trigger, NLRP3 can form an inactive spherical oligomer through interactions of its LRR domains[14–17] that, in this cage-like structure, prevent the PYD domains from forming a seed for ASC recruitment. We have previously shown that the LRR domain is not necessary for sensing triggers and inflammasome formation as well as for autoinhibition[18],

¹Department of Synthetic Biology and Immunology, National Institute of Chemistry, Ljubljana, Slovenia. ²Interdisciplinary Doctoral Study of Biomedicine, Faculty of Medicine, University of Ljubljana, Ljubljana, Slovenia. ³Institute of Cell Biology, Faculty of Medicine, University of Ljubljana, Ljubljana, Slovenia. ⁴Institute of Structural Biology, University Clinics Bonn, University of Bonn, Bonn, Germany. ⁵Centre for the Technologies of Gene and Cell Therapy, National Institute of Chemistry, Ljubljana, Slovenia. ⁶EN-FIST Centre of Excellence, Ljubljana SI-1000, Slovenia. ✉e-mail: iva.hafner@ki.si

while the sensing region was mapped to the linker-FISNA (fish-specific NACHT associated domain) segment between the PYD and NACHT domains[19]. In line with these results, active NLRP3 forms a decameric disc through its FISNA-NACHT domains, where the LRRs are not involved in oligomer formation[20].

Despite the structural and mechanistic information provided in recent years, NLRP3 remains one of the most enigmatic inflammasome-forming sensors. It is intriguing how chemically and morphologically diverse activators, originating from pathogens, environment, or self-derived molecules, can trigger NLRP3 inflammasome assembly. To date, several mechanisms have been proposed to explain this phenomenon. The decrease of intracellular $K^+$ concentration is the common denominator downstream of the majority of canonical NLRP3 instigators[21–23], yet imiquimod and related compounds drive NLRP3 inflammasome assembly independently of $K^+$ efflux[24]. A recent study demonstrated that the activation process includes inhibition of oxidative phosphorylation and an additional activating signal[25]. Moreover, different subcellular locations, cytoskeleton and organelles were previously associated with NLRP3 activation, such as mitochondria, endoplasmic reticulum (ER), Golgi apparatus (GA), endosomes, lysosomes and centrosome (microtubule organizing center, MTOC) (recently reviewed in refs. 26,27). NLRP3 can also associate with phosphatidylinositol-4-phosphate (PI4P)[28] and other negatively charged phospholipids[14] through its basic cluster (residues 127-143 in mice, 131-147 in human) and inactive NLRP3 was shown to be enriched in the GA[14,28]. Both $K^+$-dependent and independent NLRP3 triggers induce trans-Golgi network (TGN) dispersion[28] or disruption of endosome-TGN retrograde transport, which enhances PI4P accumulation and NLRP3 localization to endosomes[29–31]. MARK4 and HDAC6 were shown to facilitate NLRP3 transport to MTOC[32,33] where it can interact with NEK7[14,34]. However, NLRP3 can also associate with ER[35,36] and mitochondria[37,38] and NLRP3 inflammasome does not necessarily colocalize with MTOC[39–41].

Perplexing mechanisms underlying the cell biology of NLRP3 inflammasome activation motivated us to use a synthetic immunology approach to uncover the role of NLRP3 localization in response to diverse triggers. We demonstrate that NLRP3 does not need to be enriched at a specific cellular location; however, the association of NLRP3 with a scaffold is required for inflammasome formation. We show that engineered NLRP3 scaffolding induces NLRP3 clustering, leading to inflammasome assembly. We propose NLRP3 clustering as a universal downstream process common to different NLRP3 triggers, enabling the sensing of different types of cellular imbalance.

## Results

### Restriction to membranes of the secretory pathway supports NLRP3 activator-mediated IL-1β release and cell death

Several previous studies have suggested ER-GA compartments as important NLRP3 subcellular residence in either the dormant or activated state[14,28,35,36,42–47]. To be able to follow NLRP3 localization in live cells, we C-terminally tagged NLRP3 with a yellow fluorescent protein (YFP). When designing localization-restricted NLRP3-YFP variants (Fig. 1a, Supplementary Table 1), we aimed to tag NLRP3-YFP at the N- or C-termini to eliminate any position-specific effects, and, when possible, use more than one localization mechanism, such as the incorporation of transmembrane helices and different lipidation motifs. For example, localization of NLRP3-YFP to the cytosolic side of the ER was planned by placing transmembrane helices of cytochromes 2C1 and Cb5 to either the N- or C-terminal part of NLRP3-YFP (Supplementary Table 2). Specific localization motifs used in this study and their mechanism of action are described in Supplementary Table 1.

We previously developed a physiologically relevant system for following the effect of mutated NLRP3 variants in the inflammasome assembly, where the NLRP3 variant gene is introduced into immortalized NLRP3 knockout bone marrow-derived macrophages (NLRP3-KO iBMDMs)[18]. The expression of NLRP3-YFP variants in NLRP3-KO iBMDMs can be regulated by doxycycline concentration, which is in our experimental conditions lower than the expression of LPS-induced expression of NLRP3 in primary BMDMs, thus our results avoid overexpression that could lead to artifactual activation (Fig. 1b). While wild-type (WT) NLRP3-YFP was dispersed throughout the cytoplasm, we find that localization tags specifically enriched NLRP3 at ER, GA, or plasma membranes (Fig. 1c–e, Supplementary Fig. 1a–c). iBMDMs were primed with LPS and doxycycline to induce the expression of NLRP3 and pro-IL-1β. Afterward, the cells were challenged with the canonical NLRP3 inflammasome trigger nigericin[48]. As expected, nigericin induced notable changes in organellar structure[28,29,31], yet organelle-directed NLRP3 variants remained at compartments labeled with organelle-specific dyes, whereas non-localized WT concentrated in the perinuclear region (Fig. 1c–e, Supplementary Fig. 2a–c). Both ER-tagged variants responded to nigericin with similar levels of IL-1β release (Fig. 1f, Supplementary Fig. 3a) and cell death (Supplementary Fig. 3b) as measured by lactate dehydrogenase (LDH) activity in the cell medium. Similar results were obtained with GA-enriched variants. The response of the plasma membrane-targeted variant Lck was a bit lower than the non-localized WT NLRP3, while another plasma membrane-directed variant HRas responded similarly to the WT protein (Fig. 1f, Supplementary Fig. 3a, b).

Canonical NLRP3 activation can be triggered by many different instigators. Therefore, we were interested in whether ER-, GA- and plasma membrane-enriched variants respond to particulate triggers that cause lysosomal destabilization[49]. Indeed, the majority of ER-, GA- and plasma membrane enriched variants responded to particulate trigger silica ($SiO_2$)[49,50] with IL-1β release comparable to non-localized NLRP3-YFP (Fig. 1g, Supplementary Fig. 3c). Nigericin and particulate triggers induce $K^+$ efflux, while imiquimod activates NLRP3 inflammasome through a $K^+$ efflux independent pathway[24]. All ER-, GA- and plasma membrane-directed variants also supported imiquimod-induced IL-1β release (Fig. 1h, Supplementary Fig. 3d), demonstrating that although localization of NLRP3 was modified, it is still able to induce IL-1β release in response to $K^+$ efflux-dependent and independent triggers. Following imiquimod stimulation, organelle-directed NLRP3 variants remained at the targeted membrane, consistent with the activation with nigericin (Fig. 1c–e, Supplementary Fig. 2a–c). While the canonical NLRP3 inflammasome activation is crucial for the defense against pathogens and is suggested to participate in driving inflammation in noncommunicable diseases, mutations in the gene encoding NLRP3 can lead to cryopyrin-associated periodic syndromes (CAPS)[7]. One of these mutations is R258W (also known as R260W or R262W in human NLRP3), which leads to priming-induced NLRP3 inflammasome activation. R258 is located at the interface between the nucleotide-binding domain and the winged-helix domain of NLRP3[7]. Its exchange for the bulkier tryptophan is thought to release the autoinhibited conformation into an open, active state. Since priming is sufficient, we reasoned that the activation level of organelle-enriched R258W variants could serve as a measure of the effect of tagging and directing NLRP3 to a particular subcellular location on the downstream inflammasome processes, such as recruitment of the adaptor ASC and pro-caspase-1. We show that the R258W NLRP3 variants, enriched at ER, GA and plasma membrane, support priming-induced IL-1β release and pyroptosis to a comparable level as the non-localized variant (Fig. 1i, Supplementary Figs. 3e, f, 4a–c). The Lck variant yielded the lowest levels of IL-1β and cell death in both setups, suggesting that this tag partially restricts downstream stages of inflammasome assembly.

Our results demonstrate that the designed enrichment of NLRP3 at the ER, GA, or plasma membrane does not interfere with NLRP3 signaling instigated by canonical activators or due to CAPS-associated mutation.

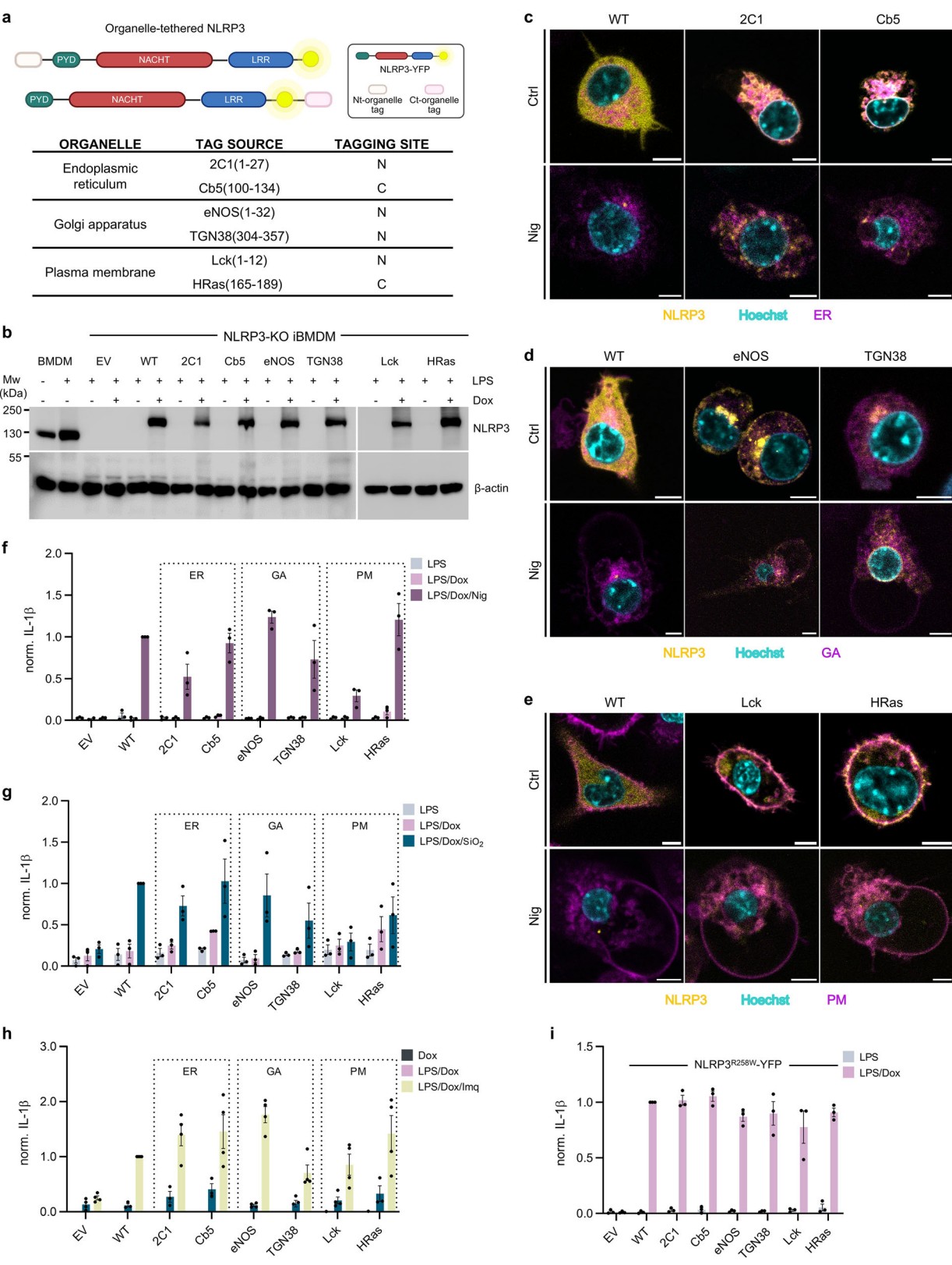

## Enrichment of NLRP3 at lysosomes, mitochondria and peroxisomes supports IL-1β release in response to canonical triggers

ROS[36] and mitochondrial DNA, particularly oxidized mitochondrial DNA[38,51,52], as well as NLRP3 association with mitochondria-localized cardiolipin[37,53], mitochondrial antiviral-signaling protein (MAVS)[54,55] or mitofusin-2[56] were shown to participate in NLRP3 inflammasome activation. Moreover, particulate triggers destabilize lysosomes to activate NLRP3[49,57–59] and NLRP3 can be located on lysosomes upon activation[60].

Using a similar strategy as before, we designed NLRP3-YFP variants to direct them to the cytosolic side of mitochondria and lysosome membranes (Fig. 2a). Although the expression levels of designed variants were different (Fig. 2b), expressed variants colocalized with Lysotracker (Fig. 2c, Supplementary Fig. 5a) or Mitotracker (Fig. 2d,

**Fig. 1 | NLRP3 enriched at ER, GA, and plasma membrane can be activated by canonical triggers. a** NLRP3 fusion proteins are designed to anchor NLRP3 to specific organelles. **b** Western blots of NLRP3-KO iBMDMs, which were primed with 100 ng/ml LPS and 1 μg/ml doxycycline (Dox) for 12 h. Primary BMDMs were plated and stimulated with LPS. **c** NLRP3-KO iBMDMs expressing the indicated NLRP3 variants (yellow), treated as in (**b**), were stained with fluorescent dyes for the endoplasmic reticulum (ER) (**c**), Golgi apparatus (GA) (**d**), or plasma membrane (PM) (**e**) (magenta) and with Hoechst 33342 for nuclei (cyan). Cells were further treated with 10 μM nigericin (Nig) for 1 h or left untreated (Ctrl). Scale bar, 5 μm. **f–h** Cells were primed as in (**b**). After the medium exchange, cells were treated with 10 μM nigericin for 1 h (**f**), 0.2 mg/ml SiO$_2$ (**g**) or 20 ug/ml imiquimod (Imq) (**h**) for

6 h. **i** NLRP3-KO iBMDMs expressing indicated NLRP3$^{R258W}$-YFP were primed as in (**b**). EV, empty vector, WT non-localized WT NLRP3. Measurements were normalized to WT cells that had been primed and subjected to canonical stimuli (**f–h**) or primed WT cells (**i**). Immunoblots (**b**) and images (**c–e**) are representative of three independent experiments. The mean ± SEM of three (**f–i**) or four ("LPS/Dox/Imq" in **h**) independent experiments is shown. The average value calculated from technical replicates within an individual experiment is presented as a single data point. One set of data for EV and WT (**h**) is identical to that in Fig. 2g, as the experiments were performed at the same time. Panel (**a**) created in Biorender. Hafner Bratkovic, I. (2025) https://BioRender.com/lwzrx04.

Supplementary Fig. 5b) dyes, respectively. Mitochondrial or lysosomal membrane localization was maintained even after the addition of canonical stimuli, nigericin and imiquimod (Fig. 2c, d, Supplementary Fig. 5c, d). In NLRP3-KO iBMDMs expressing lysosome-targeted Rheb or mitochondria-enriched Omp25 NLRP3 fusion variants, nigericin induced similar IL-1β (Fig. 2e, Supplementary Fig. 5e) and LDH release (Supplementary Fig. 5f) to cells expressing the WT NLRP3-YFP. A similar response as with nigericin was obtained when particulate instigator silica was used (Fig. 2f, Supplementary Fig. 5g). Upon priming with LPS, K⁺ efflux-independent trigger imiquimod induced comparable levels of IL-1β release from all reconstituted NLRP3-KO iBMDMs (Fig. 2g, Supplementary Fig. 5h). On the other hand, the lysosome-located Tmem192 variant was only weakly activated, likely due to poor expression level compared to WT and the other variants (Supplementary Fig. 6a, b). Tmem192 variant is also the largest (177 kDa compared to 146 kDa for WT), which may contribute to its reduced expression efficiency. Using doxycycline titration, we showed that higher expression of the Tmem192 variant leads to increased inflammasome activation, while WT reached the plateau of expression and activation at low doxycycline concentrations (Supplementary Fig. 6c, d).

As the localization of NLRP3 regarding peroxisomes has been underexplored so far, we decided to test whether NLRP3 targeted to the cytosolic face of these signaling organelles could support inflammasome assembly. We prepared constructs where PEX3 or PEX26 localization sequences were positioned to the N- or C-terminus of NLRP3-YFP, respectively (Fig. 2a). All protein variants could be detected at the protein level by western blot (Fig. 2b) and were tethered to characteristic vesicles (Fig. 2h, Supplementary Fig. 7a). Colocalization of these variants with peroxisome marker was affirmed in transfected HEK293T cells (Supplementary Fig. 7b). The PEX26 variant responded to nigericin similarly to the WT protein (Fig. 2i, Supplementary Fig. 7c). However, the PEX3 variant containing a 6 amino acid linker (PEX3$^{short}$) between the localization sequence and the NLRP3 sequence, a design also used in all other localization variants, was not responsive (Fig. 2i, Supplementary Fig. 7c). Elongation of the linker to 20 amino acids (PEX3$^{long}$) rescued the response to nigericin to the level induced by the WT protein (Fig. 2i, Supplementary Fig. 7c). To test how NLRP3 tagging with localization motifs influences downstream stages of inflammasome assembly, we again introduced the R258W mutation into these constructs (Supplementary Fig. 8a, b). In line with the response to canonical NLRP3 triggers, Rheb, Omp25 and PEX26-tagging did not interfere with priming-induced inflammasome assembly, while the response of Tmem192 and PEX3$^{short}$ variants was weaker in terms of IL-1β (Fig. 2j, Supplementary Fig. 8c) and LDH release (Supplementary Fig. 8d).

These results show that directing NLRP3 to mitochondria, lysosomes or peroxisomes does not abrogate inflammasome-mediated IL-1β release and cell death.

## Canonical NLRP3 inflammasome assembles regardless of the dominant NLRP3 subcellular location

So far, we have shown that NLRP3 variants enriched at various subcellular locations support canonical NLRP3-mediated release of IL-1β

and LDH as markers of inflammasome activation. We selected one variant to represent a particular subcellular location and followed the effect of location enrichment on several steps of inflammasome activation. We first demonstrate that macrophage cell lines, reconstituted with NLRP3 variants, respond to the priming agent (LPS) by secreting IL-6 and TNFα to a similar extent as the empty vector transduced cell line regardless of whether doxycycline, which induced NLRP3 expression, was added (Fig. 3a, b Supplementary Fig. 9a, b). Post-priming, all selected cell lines expressed similar levels of ASC, pro-caspase-1 and pro-IL-1β, further demonstrating that priming is not affected by the introduction of NLRP3 variants (Fig. 3c). We detected the presence of cleaved caspase-1 p20 subunit and IL-1β (p17) in the cell supernatant of macrophages reconstituted with NLRP3 variants while empty vector-transduced cells failed to activate caspase-1 and IL-1β in response to nigericin (Fig. 3c). In addition to IL-1β, caspase-1 also cleaves IL-18 to its mature form. Increased concentrations of IL-18 in the cell supernatant were observed for all nigericin-treated NLRP3-KO iBMDMs reconstituted with NLRP3 variants enriched at diverse subcellular locations as well as for non-localized (WT) NLRP3 but not for empty vector-transduced cells (Supplementary Fig. 9c). Furthermore, all tested NLRP3 variants induced formation of ASC specks upon treatment with nigericin to a similar level (Supplementary Fig. 9d). NLRP3 inflammasome activation in macrophages leads to pyroptosis, which makes it difficult to follow the early stages of inflammasome assembly, such as the NLRP3-ASC interaction. HEK293T cells do not express detectable amounts of inflammasome components and are a well-established system for following inflammasome assembly[15,19,33,35,43,54,61,62]. To follow NLRP3-ASC colocalization, we reconstituted HEK293T cells with fluorescent protein-labeled location-restricted NLRP3 (NLRP3-YFP) variants and ASC (ASC-mCerulean). Upon transient transfection, ASC was present in the cytoplasm and in the nucleus (Supplementary Fig. 10a), while as expected, NLRP3 variants exhibited location patterns in agreement with the respective localization tags (Fig. 3d, Supplementary Fig. 10b). Addition of nigericin induced accumulation of mCerulean fluorescence in ASC specks (Fig. 3d, Supplementary Fig. 10b), which was not the case in cells expressing ASC without NLRP3 (Supplementary Fig. 10a). To be able to follow the formation of ASC speck in inflammasome-competent cells, we introduced ASC-mCerulean into NLRP3-KO iBMDMs expressing selected NLRP3 variants (Supplementary Fig. 11a–e). In unstimulated cells, ASC was distributed throughout the cell and was barely detectable due to low expression levels. Upon activation of organelle-tethered NLRP3 variants with nigericin, ASC speck localized proximally to NLRP3 and the stained organelles, in contrast to ASC speck initiated by the WT NLRP3 control (Supplementary Fig. 11b–e).

Next, we monitored the ultrastructural changes using transmission electron microscopy. Upon nigericin treatment, cells expressing WT and location-restricted NLRP3s exhibited notable morphological changes, such as *i)* loss of ER, in some cases the ER lumen was enlarged or autophagic compartments with ER were visible; *ii)* a significant increase in GA, especially the TGN, which may have led to complete fragmentation and vesiculation of GA; *iii)* a reduction in mitochondria size, a markedly electron dense matrix,

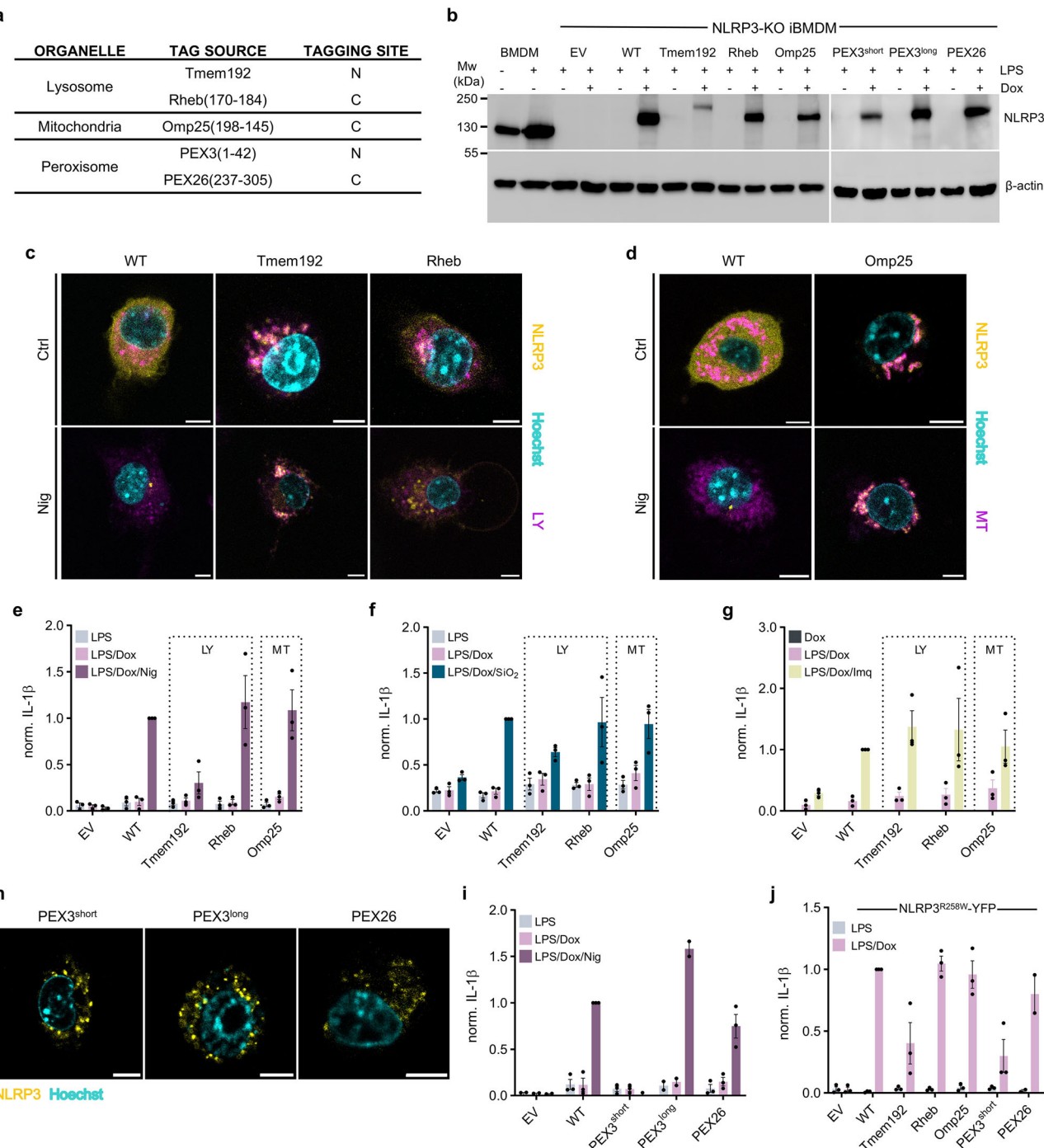

**Fig. 2 | Canonical activation of designed NLRP3 variants at lysosomal, mitochondrial, or peroxisomal membranes. a** NLRP3-YFP targeting sequences for the lysosome (LY), mitochondria (MT) and peroxisome. **b** Western blot of NLRP3-KO iBMDMs, which were primed with 100 ng/ml LPS and 1 μg/ml doxycycline (Dox) for 12 h, while primary BMDMs were primed with LPS. **c, d** Primed NLRP3-KO iBMDMs cells expressing NLRP3 variants (yellow) were stained with fluorescent dyes (magenta) for the lysosomes (**c**) or mitochondria (**d**) and nuclei (Hoechst 33342, cyan) and stimulated with 10 μM nigericin (Nig) or left unstimulated (Ctrl). Scale bar, 5 μm. **e–g, i** Cells were primed as in (**b**) and stimulated with 10 μM nigericin for 1 hour (**e, i**), 0.2 mg/ml SiO₂ (**f**) or 20 ug/ml imiquimod (Imq) (**g**) for 6 h.

**h** Peroxisome localized NLRP3-YFP variants (yellow) with stained nuclei (Hoechst 33342, cyan). Scale bar, 5 μm. **j** Constitutive activation of indicated NLRP3$^{R258W}$-YFP variants post priming. Responses were normalized to primed WT cells treated with canonical stimuli (**e–g, i**) or primed WT cells (**j**). Data in (**e–g** and **i, j**) represents the mean ± SEM of three independent experiments, except for PEX3$^{long}$ in (**i**) and PEX26 in (**j**), which are mean ± SEM of two independent experiments. The average value calculated from technical replicates within an individual experiment is presented as a single data point. One biological replicate for EV and WT is identical as in Fig. 1h. Micrographs and immunoblots are representative of three independent experiments (**b–d, h**).

alterations in the structures of the cristae and the outer mitochondrial membrane; *iv)* enlarged autophagic/ endosomal/ lysosomal compartments, particularly in TGN38, HRas and Rheb NLRP3 variants; and *v)* cytoskeleton changes (Fig. 3e). Some changes, such as increased GA, are also observed in empty vector transduced

NLRP3-KO cells (Supplementary Fig. 11f). Notably, these changes were not observed in cells expressing WT NLRP3 that were not subjected to nigericin treatment (Supplementary Fig. 11g). These observations corroborate previous studies[28,29,31] demonstrating that canonical activators such as nigericin induce significant alterations

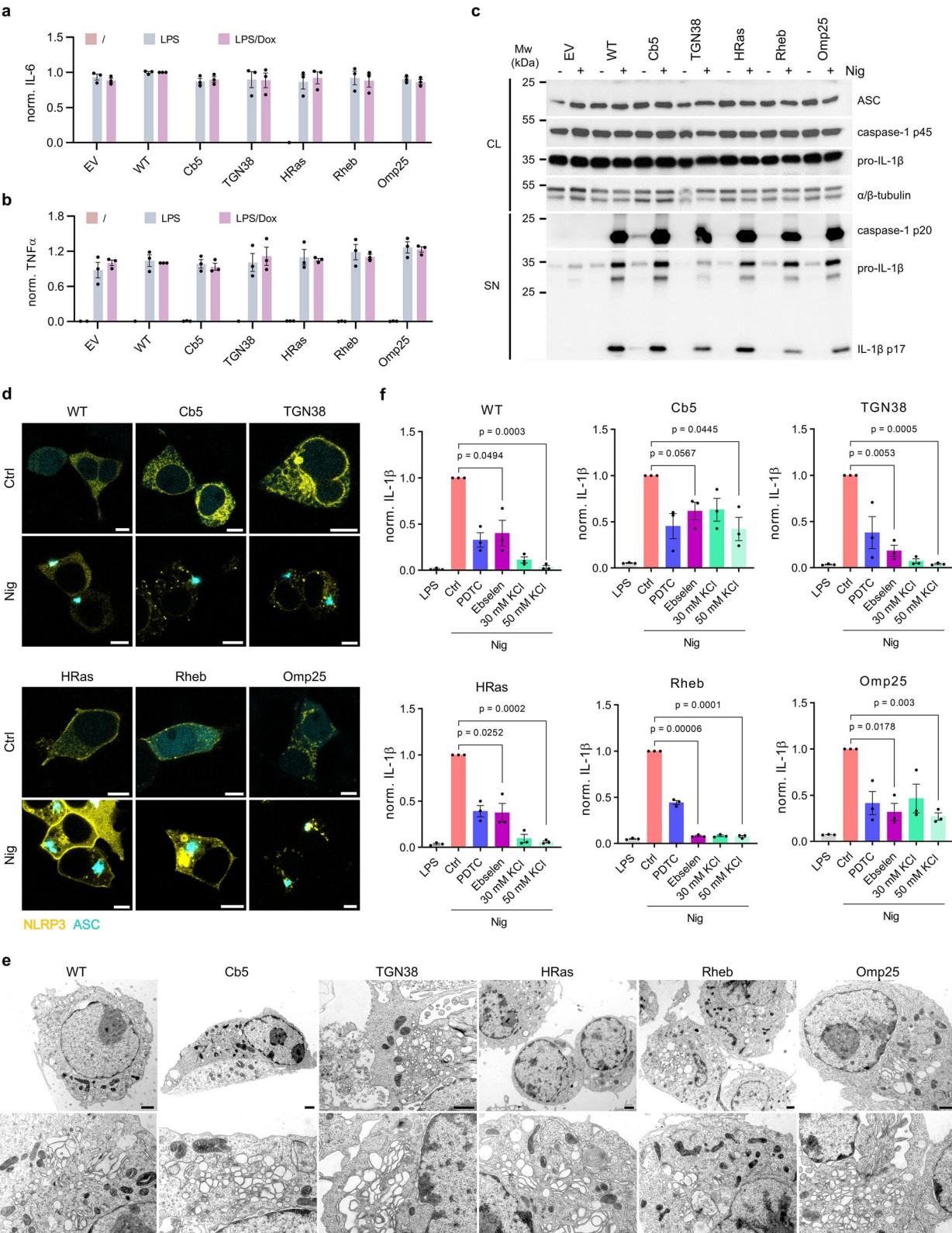

**Fig. 3 | NLRP3 activation is not restricted to specific membranes. a, b** Levels of IL-6 and TNFα from primed and doxycycline (Dox)-treated NLRP3-KO iBMDMs were normalized to primed WT cells. **c** Cell lysates (CL) of primed and nigericin (Nig)-treated cells were analyzed for expression of ASC, pro-caspase-1, pro-IL-1β and cell supernatants (SN) for cleaved caspase-1 and IL-1β. **d** HEK293T expressing ASC-mCerulean (cyan) and indicated NLRP3-YFP variants (yellow) were treated with or without nigericin. Scale bar, 5 μm. The representative micrographs of nigericin-treated NLRP3-KO iBMDMs show ultrastructural changes, especially in the GA with enlargement of the TGN and vesiculation of the GA. The mitochondria have an

electron-dense matrix and are smaller. The ER is less pronounced and the autophagic/endosomal/lysosomal degradation compartments are enlarged. Scale bar, 1 μm. **f** After priming, KCl or ROS inhibitors were added for 30 minutes, and nigericin was added for 1 h. Bars in (**a**, **b** and **f**) represent the mean ± SEM of three independent experiments. The average value calculated from technical replicates within an individual experiment is presented as a single data point. Immunoblots (**c**) and micrographs (**d**, **e**) are representative of three independent experiments. Two-tailed unpaired *t*-test with Welch correction was used for the comparison of populations with different variances (**f**).

in the cell ultrastructure that are not a consequence of NLRP3 inflammasome activation.

We therefore demonstrated that NLRP3 variants enriched at various subcellular locations facilitate all stages of NLRP3 inflammasome activation, regardless of their predominant localization, suggesting that NLRP3 location is not governing the inflammasome assembly.

## Nigericin-stimulated inflammasome activation of localized variants depends on K[+] efflux

As many of the selected constructs enrich NLRP3 at membranes of the secretory pathway, we intended to observe whether anterograde and retrograde trafficking modify inflammasome activation. If a specific subcellular location would be required or preferred for inflammasome assembly, chemical inhibitors should differently affect inflammasome activation by NLRP3 variants located at diverse organelles. We first showed that the NLRP3 inhibitor MCC950[61–63] and the deubiquitinase inhibitor G5[64] similarly suppressed nigericin-induced release of IL-1β supported by different NLRP3 variants (Supplementary Fig. 12a–i). Further, we tested several known inhibitors targeting anterograde trafficking: golgicide A, AG1478 and brefeldin A (reviewed in ref. 65). The effect of AG1478 was similar for all tested variants, while golgicide A and brefeldin B had varying levels of inhibition. We should note that golgicide A and brefeldin A were previously shown to affect IL-1β secretion, but acted downstream of NLRP3 as IL-1β secretion due to NLRP1 inflammasome activation was similarly affected[66]. Vacuolin-1, the inhibitor of exocytic fusion of lysosomes with the plasma membrane[67], did not affect or only mildly promoted nigericin-induced IL-1β secretion. Furthermore, we tested four compounds that abrogate retrograde trafficking. Retro-2 and tyrphostin A23 had mild to moderate effects on the secretion of IL-1β, pitstop 2 strongly inhibited IL-1β secretion, whereas dynasore had various effects on cells expressing different mutants. The compounds did not induce IL-1β secretion from nigericin-treated empty-vector transduced NLRP3-iBMDMs (Supplementary Fig. 12j) or pro-IL-1β levels (Supplementary Fig. 12k) as expected since they were added after the priming step. Despite some differences, chemical inhibitors of anterograde and retrograde trafficking did not point to any preferred NLRP3 subcellular localization as the prerequisite for NLRP3 inflammasome assembly.

Nigericin and the majority of canonical NLRP3 instigators induce a decrease in the intracellular concentration of K[+], leading to NLRP3 inflammasome activation[21–23]. K[+] efflux was shown to promote NLRP3 conformational change into an open state[19]. We were interested in whether localized NLRP3 variants remain sensitive to K[+] efflux. Therefore, we preincubated primed cells in media with added KCl at concentrations that were previously shown not to affect other cellular processes, such as priming, or inhibit imiquimod-induced NLRP3 activation[24]. As expected, the increased extracellular KCl dampened IL-1β secretion of WT downstream of nigericin but not imiquimod (Fig. 3f, Supplementary Fig. 13). Prevention of K[+] efflux also decreased inflammasome activation supported by organelle-localized NLRP3 variants (Fig. 3f, Supplementary Fig. 13). While imiquimod was shown to be independent of K[+] efflux, chemical inhibitors that suppress ROS were shown to dampen imiquimod-induced NLRP3 inflammasome activation[24]. As in the case of WT NLRP3, ebselen and pyrrolidine dithiocarbamate (PDTC) suppressed secretion of IL-1β triggered by nigericin (Fig. 3f) and imiquimod (Supplementary Fig. 13) from cells harboring location-restricted NLRP3 variants. Therefore, the dependence of organelle-tethered NLRP3 and WT NLRP3 inflammasome activation on canonical activators as well as on K[+] efflux suggests a shared activation pathway.

## Association of NLRP3 with membranes facilitates trigger-induced inflammasome assembly independently of the basic segment

The linker-FISNA segment between the PYD and the NACHT domains of NLRP3 is the region that has been recently proposed to sense changes, caused by different canonical NLRP3 triggers[19]. This region also contains a positively charged segment (Fig. 4a) that enables association with PI4P and is crucial for mouse NLRP3 response[28–30]. Since the designed variants were already enriched at particular subcellular membranes, we were interested, whether this basic segment and thus the interaction with PI4P was still necessary for inflammasome assembly supported by designed membrane-enriched NLRP3. We prepared several constructs previously shown to tether NLRP3 to the cytosolic side of ER (Cb5), GA (TGN38), plasma membrane (HRas), lysosomes (Rheb) and mitochondria (Omp25) by mutating four lysine residues (K127-K130) to alanine (4xA). Upon introduction into NLRP3-KO iBMDMs and addition of doxycycline, the 4xA mutants were expressed to comparable levels as their nonmutated counterparts (Fig. 4b). WT NLRP3 and 4xA mutant were cytosolic (Figs. 1c, 4c) in the absence of triggers and location-restricted 4xA variants were associated with their targeted membranes (Fig. 4d–h, Supplementary Fig. 14). Consistent with the report from Chen and Chen[28], mutant NLRP3 4xA failed to induce IL-1β release upon activation with nigericin, unlike nonmutated WT NLRP3 (Fig. 4c). Similarly, NLRP3 4xA did not respond to silica and imiquimod treatment (Fig. 4c). Surprisingly, despite the 4xA mutation, membrane-enriched variants were still able to form puncta when exposed to nigericin (Fig. 4d–h, Supplementary Fig. 14), which correlated with their ability to mediate IL-1β release (Fig. 4d–h). Membrane-tethered NLRP3 variants were able to support IL-1β secretion upon silica and imiquimod treatment, whereas cytosolic 4xA failed (Fig. 4c–h). Notably, IL-1β release was comparable from both membrane-enriched non-mutated NLRP3 and its alanine mutant counterpart (Fig. 4d–h). The observation that cytosolic NLRP3 lacking the charged segment is dysfunctional, while location-enriched 4xA variants facilitate inflammasome activation, strengthens our conclusion that different subcellular locations of NLRP3 facilitate inflammasome activation. This effect could not be attributed to a potentially small fraction of NLRP3 exhibiting incomplete membrane restriction.

These results demonstrate that constitutive or inducible binding of NLRP3 to membranes is important but not sufficient for inflammasome activation.

## Attachment of NLRP3 to different scaffolds facilitates inflammasome assembly

We have demonstrated that NLRP3 variants designed to primarily reside at the cytosolic faces of ER, GA, plasma membrane, lysosome, peroxisome, and mitochondria respond to canonical inflammasome activators comparably to non-localized WT NLRP3 and were not constitutively active. Interestingly, two variants (Fig. 5a) that were designed to enrich NLRP3 at centrosome (PACT) and cis/medial GA (giantin, GOLGB1) induced IL-1β secretion (Fig. 5b), pore formation (as judged by propidium iodide uptake, Fig. 5c), and cell death (LDH release, Fig. 5d) in the absence of canonical NLRP3 trigger. This observation is in line with the CAPS mutant R258W phenotype, while WT NLRP3 needs the second signal for its activation (Fig. 5b–d). However, the addition of nigericin further enhanced the activation of GOLGB1 variant (Fig. 5b–d). Priming-induced activation of PACT and GOLGB1 variants was additionally confirmed by western blot as the processed subunits of caspase-1 and IL-1β were present in the cell supernatant of primed-only cells (Fig. 5e). Priming induced by LPS treatment was similar for all cell lines based on the expression of pro-IL-1β (Fig. 5e) and secretion of LPS-induced proinflammatory cytokines IL-6 (Fig. 5f) and TNFα (Fig. 5g). Trigger-independent activity of PACT and GOLGB1 could not be attributed to the tag-induced oligomerization as variants with longer linker also supported IL-1β release in the absence of the second signal (Supplementary Fig. 15a, b). PACT, GOLGB1 and CAPS-related R258W mutant and their counterparts where the basic segment (K127-K130) was mutated (4xA variants) (Supplementary Fig. 15c) were primed overnight, after which the medium was exchanged, and cells were challenged with NLRP3

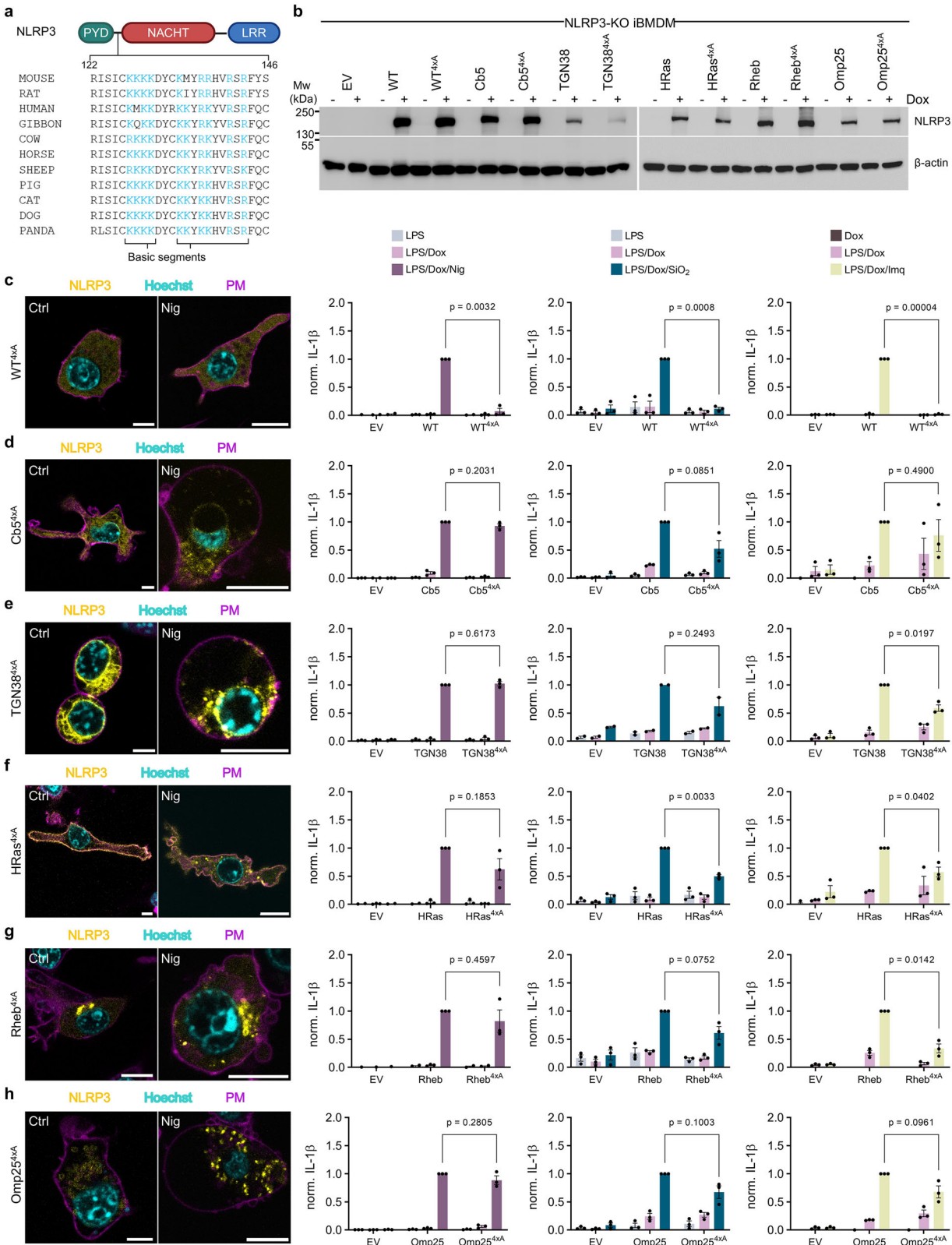

activators. We observed a high activation of the 4xA mutants even in the absence of the NLRP3 triggers nigericin, silica and imiquimod (Fig. 5h–j), suggesting that as in the case of the CAPS-related R258W variant, the basic segment and consequently association with the PI4P are redundant for activation of PACT- and GOLGB1-tagged NLRP3. The enrichment of priming only-activated variants at different subcellular locations, either on the membrane or MTOC, both serving as a lipid or protein scaffold, suggests that membrane association is not crucial for inflammasome assembly, but can be compensated with a scaffold of a different type.

## Membrane and protein scaffolds promote NLRP3 clustering

In CAPS mutants, destabilization of the autoinhibited conformation is proposed to mediate inflammasome activation[68]. We were intrigued by

**Fig. 4 | NLRP3 interaction with membranes supports inflammasome activation.**
**a** Conserved basic segments within the linker and FISNA region of NLRP3 homologs. **b** Immunoblot of non-localized and organelle-tethered NLRP3-YFP with respective NLRP3 mutants K127A/K128A/K129A/K130A (4xA). **c**–**h** Confocal images of NLRP3-KO iBMDMs stably transduced with organelle-tethered NLRP3-YFP (yellow) variants 12 h post priming with or without nigericin stimulation (Nig), stained for nuclei (Hoechst 33342, cyan) and plasma membrane (Cholera Toxin Subunit B, CTB, magenta). Scale bar, 10 μm. Following priming, 10 μM nigericin, 0.2 mg/ml $SiO_2$ or 20 ug/ml imiquimod (Imq) were added to cells, respectively, as indicated in the respective graph columns with legends above. Measurements of IL-1β were normalized to each respective non-mutated, organelle-enriched NLRP3-YFP: WT (**c**), Cb5 (**d**), TGN38 (**e**), HRas (**f**), Rheb (**g**) or Omp25 (**h**) treated with canonical stimuli: nigericin, $SiO_2$ or imiquimod, respectively. EV, empty vector. Data represents the mean ± SEM of three independent experiments, except for TGN38, treated with $SiO_2$ in (**e**), which is the mean ± SEM of two independent experiments. The average value calculated from technical replicates within an individual experiment is presented as a single data point. P values were calculated using two-tailed unpaired *t*-test with Welch's correction (**c**–**h**). Western blots (**b**) and microscopic images (**c**–**h**) are representative of three independent experiments. Panel (**a**) created in Biorender. Hafner Bratkovic, I. (2025) https://BioRender.com/99kdc99.

how two NLRP3 variants, one tagged at the C-terminus and the other at the N-terminus, phenocopied R258W without additional modifications in the NLRP3 sequence (Fig. 5b–d). Upon priming, PACT-tagged NLRP3 concentrated to the spot corresponding to the MTOC, as judged by colocalization with γ-tubulin (Fig. 6a, Supplementary Fig. 15d). GOLGB1-tagged NLRP3, on the other hand, formed multiple spots in the cell (Fig. 6a, Supplementary Fig. 15e). In fact, the majority of cells expressing those variants contained NLRP3 puncta (Fig. 6a, Supplementary Fig. 15f), similar to those in activated cells (Figs. 1c–e, 2c, d and 4d–h).

The constitutively active PACT variant, enriched at MTOC, suggests that the scaffold enabling clustering of NLRP3 and subsequent activation does not need to be a lipid membrane. We sought to determine whether forced oligomerization of NLRP3 through protein oligomerization domains can be sufficient to induce activation without the addition of activators. TAR DNA-binding protein 43 (TDP43) forms liquid condensates by binding to cytosolic RNA[69]. We fused TDP43 to NLRP3-YFP at the C-terminus (Fig. 6b) to avoid possible direct oligomerization of NLRP3[PYD] domains. TDP43 fusion enabled oligomerization of NLRP3 (Fig. 6c) and induced multiple NLRP3 clusters per cell regardless of the presence of the basic segment (K127-130) in NLRP3 (Fig. 6d, Supplementary Fig. 16a). We were able to capture some cells that contained clusters but did not yet form GSDMD pores (as indicated by the lack of propidium iodide intercalation), demonstrating that clustering is not a consequence of cell death, but rather actively promotes inflammasome assembly (Fig. 6d, Supplementary Fig. 16a). In line with previous results regarding PACT variant, fusion with TDP43 induced potent constitutive activity that could not be further potentiated by nigericin and was independent of the presence of the basic segment (Fig. 6e, Supplementary Fig. 16b, c).

This demonstrates that membrane or protein scaffolds facilitate either trigger-induced or spontaneous NLRP3 clustering that promotes NLRP3 inflammasome activation.

## Clustering of NLRP3 relaxes the inactive NLRP3 NACHT conformation

The inactive form of NLRP3 is stabilized by LRR-domain-mediated spherical assemblies[14–16]. To form the active disc, disruption of LRR-LRR interactions as well as important conformational changes within the NACHT domain are required to enable ADP for ATP exchange and NACHT oligomer formation[20].

To determine how membrane-tethered NLRP3 is regulated at the molecular level and whether the activity upon priming of PACT-, GOLGB1- or TDP43-tagged variants results from disassembly of inactive LRR-domain mediated cages, we prepared GA-localized eNOS-tagged variants that cannot form inactive cages. We have previously shown that the minimal active NLRP3 variant (1-686) lacking the LRR domain was not constitutively active despite not being able to form inactive cages[18]. In addition, we prepared GA-localized NLRP3 F785A/D786R/F810A variant (eNOS-Δcage), in which the intermolecular back-to-back LRR interaction facilitating the formation of the inactive cage is disrupted[15]. Localization of eNOS variants was not affected by the changes in the NLRP3 sequence (Supplementary Fig. 16d, e). Cells

containing those mutants were activated with canonical stimuli nigericin (Supplementary Fig. 16f) and imiquimod (Supplementary Fig. 16g), and exhibited no constitutive activity, suggesting that in the absence of triggers, the GA-enriched trigger-responsive variant is in the inactive, most likely closed NACHT domain conformation.

PACT, GOLGB1 and TDP43 tags by enriching NLRP3 on different scaffolds might induce an arrangement of NLRP3 PYD domains that is able to recruit the adaptor ASC and thus initiate inflammasome assembly in the absence of inflammasome instigators. We have previously shown that the NLRP3[PYD] domain when fused to the trimerization domain foldon induces robust trigger-independent inflammasome activation[70]. We used a panel of well-characterized NLRP3 inhibitors to get an insight into the mechanism of action of these constitutively active variants. As mentioned before, G5 is an indirect NLRP3 inhibitor, while MCC950 is a specific NLRP3 inhibitor that acts by tethering NLRP3 subdomains in the inactive conformation[15,16,19,62,63,71]. Oridonin inhibits NLRP3 activation by covalently binding to Cys279[72], whereas RRx-001 covalently modifies Cys409[73] and Cy-09 is supposed to bind to the Walker A motif[74]. All inhibitors suppressed IL-1β release in both R258W mutant (Fig. 6f) and nigericin-treated WT cells in a dose-dependent manner (Supplementary Fig. 16h). On the other hand, none of the inhibitors affected NLRP3[PYD]-foldon-mediated IL-1β secretion, since compounds do not target NLRP3[PYD] domain or its association with ASC[PYD] (Fig. 6g). These results also demonstrate that with the concentrations used, these compounds do not affect either priming or downstream inflammasome activation stages such as caspase-1 activation or activity. Unexpectedly, while oridonin, RRx-001, Cy-09 and G5 dose-dependently dampened activation of GOLGB1, PACT, and TDP43 NLRP3 variants, MCC950 had no effect on the activation of PACT and TDP43 variants and showed only a mild effect on GOLGB1 without a clear concentration-dependent suppression (Fig. 6h–j). NLRP3-specific inhibitors do not affect NLRP3[PYD]-foldon-mediated inflammasome activation, but effectively suppress the activity of PACT, GOLGB1 and TDP43 variants, demonstrating that NLRP3 clustering-supported activation does not originate from tag-induced oligomerization of NLRP3[PYD] domains. This is further supported by a panel of NLRP3-TDP43 mutants targeting the conserved Walker A motif, which disrupt NLRP3's ability to bind ATP, previously shown to be critical for NLRP3 oligomerization and activation[15,20,75,76]. We prepared two Walker A mutants, in which either three residues (G227, K228, T229) were mutated to alanine (3xA) or K228, responsible for coordinating the β-phosphate of ATP, was mutated to alanine (K228A). Similar to WT, where nigericin-induced activation strictly relied on ATP binding, neither TDP43[K228A] nor TDP43[3xA] were able to induce IL-1β secretion, either constitutively or even after addition of nigericin (Fig. 6k, Supplementary Fig. 16i), demonstrating that clustered NLRP3 variants similarly to WT NLRP3 require ATP binding for inflammasome formation.

The inability of MCC950 to inhibit PACT, GOLGB1 and TDP43 variants was surprising. MCC950 is known to hinder the conformational change required for activation by holding the subdomains of the NLRP3 NACHT together; yet, it is ineffective on several CAPS mutations that either disrupt its binding sites or strongly affect the autoinhibited

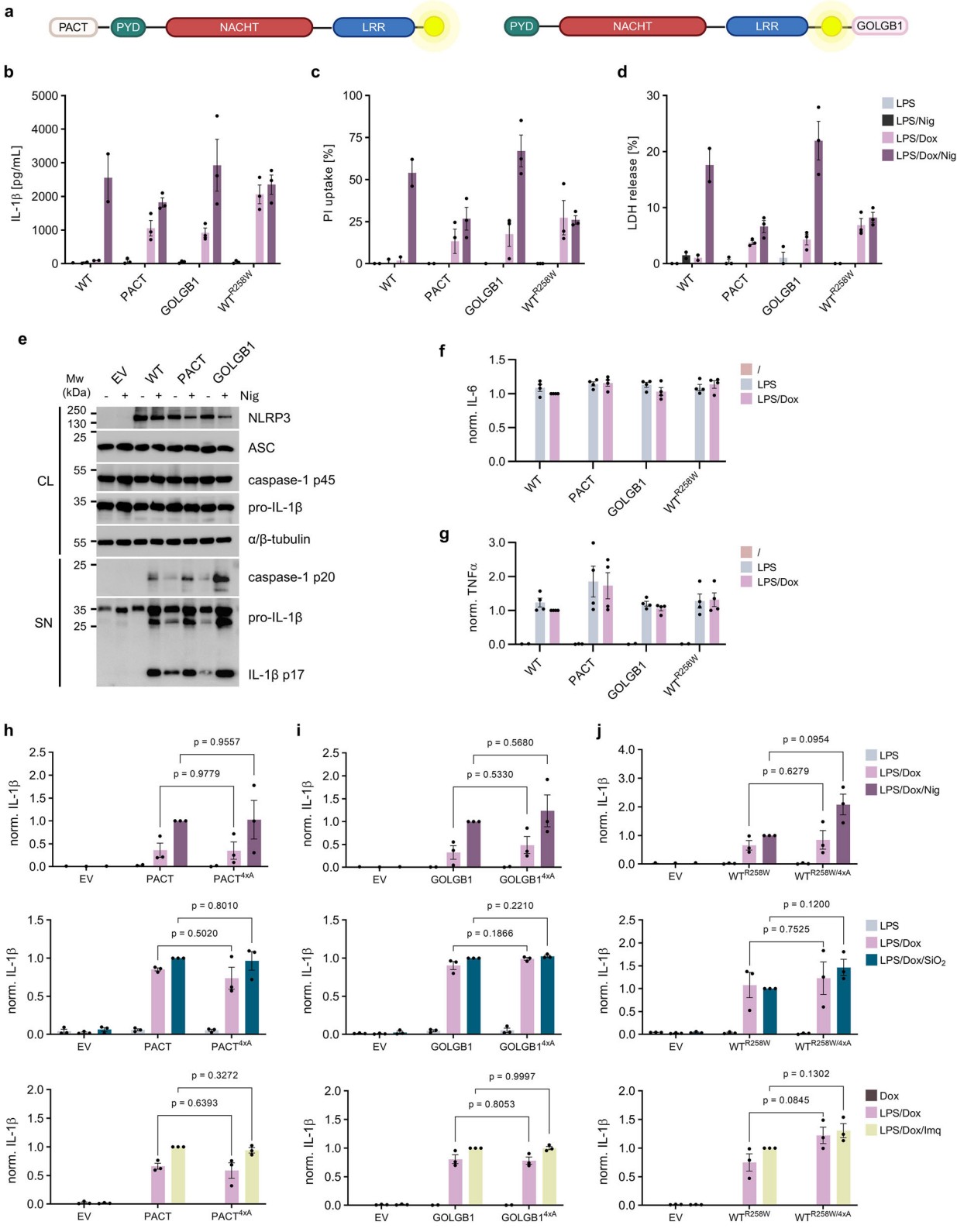

state of the AAA+ ATPase[15,61–63,68,71]. Structural transitions within NLRP3 are rather complex, involving the formation of inactive LRR-domain-based cages[14–16] that need to dissolve and a conformational transition of the NACHT domain to form the NLRP3 active disc[20]. Given that no mutations were introduced into the NLRP3 sequence in our variants, we propose that clustering of NLRP3 on different scaffolds facilitates rearrangement of the inactive NLRP3 structure, which prevents

binding of MCC950 and helps to unlock the inhibited NLRP3 conformation, endorsing active NACHT-based oligomer formation (Fig. 7).

## Discussion

One of the major enigmas connected to the NLRP3 inflammasome is how a single immune sensor is able to detect so many unrelated threats. NLRP3 activators were shown to cause ion imbalances, such as

**Fig. 5 | Constitutively active PACT and GOLGB1-tagged NLRP3 variants. a** Design of variants at MTOC (PACT, aa 3699–3790) and Golgi apparatus (GOLGB1, aa 3131-3259). **b–d** NLRP3-KO iBMDMs were primed and after 12 hours, nigericin (Nig) was added. Maturation of IL-1β (**b**) and GSDMD pore formation by propidium iodide (PI) uptake (**c**), and cell death by LDH release (**d**) were measured. As a control, NLRP3-KO iBMDMs encoding NLRP3-YFP (WT) were treated with only LPS and Nig. **e** Lysates (CL) from primed and Nig treated cells were immunoblotted for NLRP3, ASC, caspase-1, IL-1β and supernatants (SN) for mature caspase-1 and IL-1β. Representative of three independent experiments. Supernatants from (**b**) were analyzed for IL-6 (**f**) and TNFα (**g**) and normalized to primed ("LPS/Dox") WT-expressing cells. **h–j** Post-priming, nigericin, SiO$_2$ or imiquimod (Imq) were added to cells, respectively. Release of IL-1β was followed and normalized to primed and Nig-treated cells expressing PACT (**h**), GOLGB1 (**i**) or WT$^{R258W}$ (**j**). Data represents the mean ± SEM of two (WT in **b–d** and "/" for WT$^{R258W}$ in **f**, **g**), three (**b–d**, **h–j**) or four (**f**, **g**) or independent experiments. The average value calculated from technical replicates within an individual experiment is presented as a single data point. Two-tailed unpaired t-test ("LPS/Dox" in **h–j**, except for middle panel in **h**) and an unpaired t-test with Welch's correction ("LPS/Dox/Trigger" in **h–j**) were employed. Panel (**a**) created in Biorender. Hafner Bratkovic, I. (2025) https://BioRender.com/3a80hvb.

K$^+$ efflux, oxidative stress and organelle dysfunction, endorsing NLRP3 as an important detector of alterations in cell homeostasis[77,78].

Seminal studies described a crucial role of NLRP3 trafficking during different stages of activation to eventually reach the location where NLRP3 forms the inflammasome (reviewed in refs. [26,27,79]). For instance, in the dormant state, NLRP3 was reported to localize at ER, and after activation forms the inflammasome at mitochondria-associated membranes[36,42]. Additionally, NLRP3 interacts with the SREBP2/SCAP complex that shuttles it from the ER to the GA[43]. Arumugam et al. describe NLRP3 recruitment to mitochondria and later via GA to the TGN[46]. In the primed state, NLRP3 is enriched at the GA[28,47] and MARK4 and HDAC6 promote NLRP3 transport to MTOC[32,33], thereby enabling its interaction with NEK7[14,34]. Some studies described the pathway as specific for a particular trigger or animal species, while others demonstrated the pathway common to diverse activators. In our study, we focused on murine NLRP3 and performed most of the experiments in mouse iBMDMs, but we addressed the question of whether different triggers impose distinct restrictions. We were surprised to reveal a highly promiscuous nature of NLRP3 regarding the subcellular location that permits the functional inflammasome assembly. We determined that NLRP3, when enriched at the cytosolic faces of mitochondria, ER, GA and lysosomes, supports inflammasome assembly downstream of activators with different modes of activation, such as small molecule and particulate species that induce K$^+$ efflux, as well as the K$^+$-independent instigator imiquimod. While we corroborate the above studies in terms of the propriety of highlighted NLRP3 locations for inflammasome activation, we challenge the concept of the exclusive NLRP3 inflammasome-forming location, such as mitochondria or MTOC, which have also been questioned in recent studies[39–41]. Using the human system, Mateo-Tórtola and colleagues describe two pathways, MTOC-dependent and independent, that proceed simultaneously within human cells in response to nigericin[39]. Moreover, we demonstrate that even restriction of NLRP3 to membranes not often associated with NLRP3 inflammasome formation, such as of peroxisomes and plasma membrane, support inflammasome activation, suggesting that a particular cytoplasmic location of NLRP3 is not the limiting factor in response to various activators.

We show here that the forced association of mouse NLRP3 with different membranes is not sufficient for spontaneous inflammasome assembly. While nigericin activation of membrane-tethered variants depends on K$^+$ efflux, in contrast to WT murine NLRP3, membrane-tethered variants were activated in the absence of the positively charged segment in the linker region. This observation is in line with studies showing that NLRP3 association with PI4P via basic cluster or PI4P-binding domain primes NLRP3[28,47]. NLRP3 activators induce major changes within the cell that lead to inflammasome formation, which is consistent with our ultrastructural analysis of nigericin-treated mouse macrophages. Chen and Chen proposed that dispersion of TGN changes the curvature of dispersed TGN vesicles, which facilitates NLRP3 oligomerization[28], however, with additional markers, two groups identified those vesicles to be of endosomal origin[29,30]. Recruitment of NLRP3 to endosomes is additionally supported by a recent proteomic study[31]. Particulate triggers cause the enlargement of lysosomes and their

destabilization[49] as well as plasma membrane leakage[80]. In terms of membrane association, not only can NLRP3 bind PI4P[28], but mouse NLRP3 was also shown to associate with other phosphoinositides and phosphatidic acid[14] and cardiolipin[16,37,53], suggesting that other (non-PI4P enriched) membranes could be involved in activation[81]. Interestingly, isolated human NLRP3, where the KMKK motif was shown to be dispensable for inflammasome activation[19], failed to bind all mentioned lipids apart from phosphatidic acid and PIP3[39]. Such differences can contribute to the observed interspecies variations in inflammasome responses. Moreover, several recent reports uncovered that palmitoylation of NLRP3 promotes inflammasome activation[82–85], which further supports the idea that the association of NLRP3 with various membranes enables NLRP3 inflammasome responses. It is likely that different ultrastructural changes, such as membrane curvature and enrichment in phosphoinositides associated with diverse organelles, drive NLRP3 clustering, which we identify as crucial for inflammasome formation while being agnostic to the precise cellular location.

We were not able to prepare an MTOC-enriched trigger-responsive variant. However, the PACT variant, active in the absence of instigators, revealed that a membranous scaffold could be substituted for a protein-based scaffold, which we further demonstrated by fusing NLRP3 to TDP43, a protein that forms membrane-less organelles associated with neurodegeneration. Besides the lipids mentioned above, different NLRP3 ligands can in principle serve as a scaffold for binding NLRP3 and facilitate its clustering. Oxidized mitochondrial DNA[51,52] was shown to be crucial for NLRP3 activation and NLRP3 can dock to mitochondria after the addition of various canonical, non-canonical, or alternative activators via binding to oligomerized VDAC/mtDNA[38]. RNA viruses induce association of NLRP3 with proteins, present at the outer mitochondrial membrane, MAVS[54,55] or mitofusin-2[56]. On the other hand, the dsDNA virus HSV-1 engages STING to recruit NLRP3 to the ER[35]. Some NLRP3 ligands were even shown to have dual recruiting roles. HDAC6 facilitates microtubule-based NLRP3 transport to MTOC[32] but also mediates the interaction and positioning of NLRP3 with Lamtor1 on lysosomes[60]. However, NLRP3 activation will not be facilitated by any negatively charged scaffold. This is nicely demonstrated by Samir et al. who showed that DDX3 promotes NLRP3 activation through interaction with the NACHT domain[86], but will fail to interact with NLRP3 and facilitate inflammasome assembly when localized in stress granules[86]. While these studies contribute to the complexity of the NLRP3 interactome and may help to explain how NLRP3 can be activated ubiquitously, our study reveals the necessity for spatial concentration of NLRP3. NLRP3 may be able to bind to various scaffolds, such as a membrane, protein or nucleic acid as illustrated in Fig. 7. Those scaffolds might need further rearrangements distinct from those in homeostatic conditions to induce NLRP3 clustering and inflammasome formation. Using NACHT domain binding NLRP3 inhibitors we were able to identify that clustering of NLRP3 most likely destabilizes intramolecular interactions that keep the NACHT domain in the inactive conformation. We hypothesize that this could be a result of increased local concentration of NLRP3 and/or orientation of NLRP3 molecules that could override inhibitory interactions.

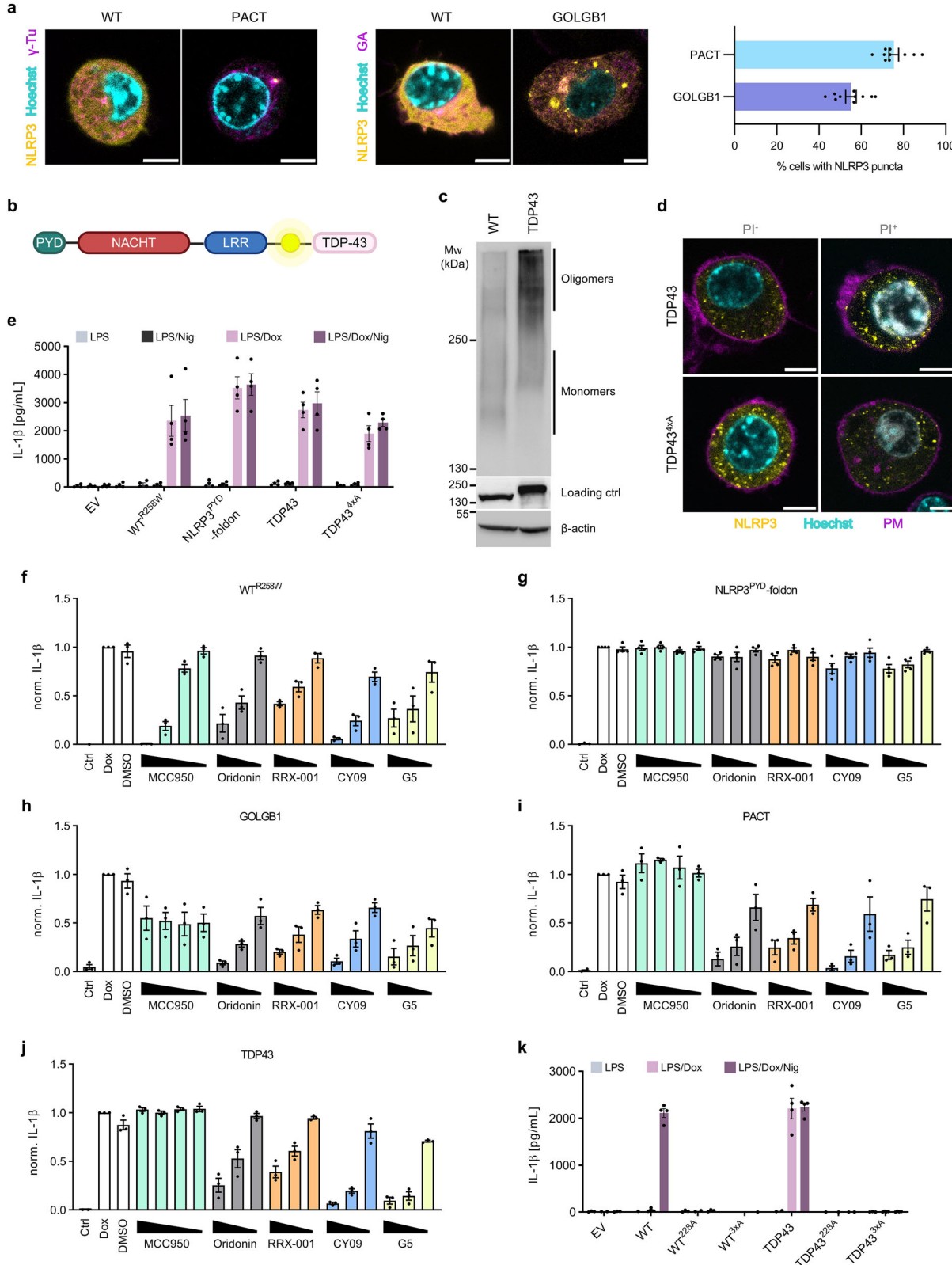

Using a molecular immunology approach, we show that NLRP3 has a wide activation space and can together with previously described diverse binding partners provide a prompt response to a variety of threats. We propose a mechanism of activation where different instigators or injuries promote cellular disbalance which leads to clustering of NLRP3 on diverse cellular surfaces leading to the rearrangement of NLRP3 and subsequent active disc formation (Fig. 7). Clustering of NLRP3 thus presents a key and distinct stage in the activation pathway. The universality of NLRP3 response to diverse types of cellular insults explains why NLRP3 is regulated at so many different levels to impede unnecessary activation, while also positioning NLRP3 as the cell's master danger sensor.

**Fig. 6 | Different types of scaffolds induce clustering of NLRP3. a** Primed cells expressing NLRP3 variants (yellow) were stained for γ-tubulin (γ-Tu) or Golgi apparatus (GA), respectively, depicted in magenta, and nuclei (Hoechst 33342, cyan). Scale bar, 5 μm. For quantification of NLRP3 puncta, five random frames were recorded for each cell line. The number of cells with puncta was divided by the number of cells expressing NLRP3 within each frame. **b** Design of NLRP3-TDP43 fusion protein. **c** Native PAGE of primed NLRP3-expressing cells. **d** TDP43 NLRP3 or TDP43 NLRP3 4xA mutant (NLRP3$^{K127A/K128A/K129A/K130A}$) expressing cells were stained with propidium iodide (PI, grey), Hoechst 33342 (cyan) and Cholera toxin subunit B (PM, magenta). Scale bar, 5 μm. **e, k** Post-priming with LPS and doxycycline (Dox), NLRP3-KO iBMDMs were activated with nigericin (Nig). **f–j** NLRP3-KO iBMDMs were stimulated with LPS and Dox after a 30-min pre-incubation with inhibitors or DMSO, except for Ctrl, where only LPS was added. Measurements of IL-1β were normalized to "LPS/Dox"- treated WT$^{R258W}$ (**f**), NLRP3$^{PYD}$-foldon (**g**), GOLGB1 (**h**), PACT (**i**) or TDP43 (**j**). Data are mean ± SEM of two (**a** right), three (**f, h, i, j**, TDP43$^{228A}$ in **k**) or four (**g, e, k**) independent experiments. Micrographs (**a, d**) and western blot (**c**) are representative of three independent experiments. Panel (**b**) created in Biorender. Hafner Bratkovic, I. (2025) https://BioRender.com/6wlxr0t.

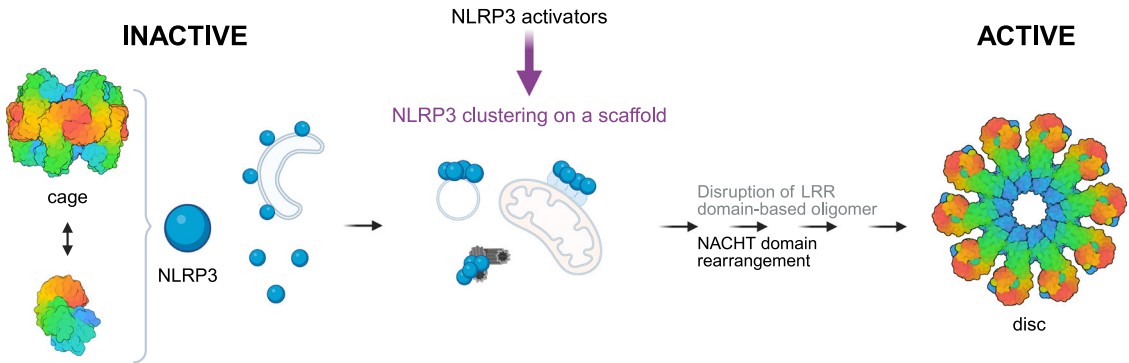

**Fig. 7 | Scaffold-facilitated NLRP3 clustering promotes inflammasome assembly.** Inactive NLRP3 species exist as cage-like structures and monomers, residing either in the cytosol or associated with different membranes. NLRP3 instigators induce changes in the cell ultrastructure leading to the formation/rearrangement of various scaffolds that enable NLRP3 clustering, thereby facilitating further steps that lead to active disc formation. Models of the NLRP3 are based on the structures with PDB IDs 7PZC (cage)[15], 4KXF (inactive NLRC4)[94] and 8EJ4 (active disc)[20]. Created in Biorender. Hafner Bratkovic, I. (2025) https://BioRender.com/wzlmple.

## Methods

### Plasmid construction and preparation

To prepare NLRP3 localized variants, the construct encoding mouse NLRP3 tagged with YFP including linker sequence (amino acid sequence SGPGSG)[18] was amplified with primers and ligated with the relevant localization sequence into the pRETROX TRE3G plasmid (Clontech) at BamHI/EcoRI or BamHI/NotI sites. Localization sequences were amplified from plasmids encoding Hras (a gift from D. Sabatini, Addgene 26637[87]), Tmem192 (a gift from D. Sabatini, Addgene 102930[87]), Rheb (a gift from D. Sabatini, Addgene 26634[87]), TGN38 (a gift from J. Lippincott-Schwartz, Addgene 128148), GOLGB1 (a gift from D. Gadella, Addgene 67903[88]), Omp25 (a gift from D. Sabatini, Addgene 26638[87]), PEX3 (a gift from J. Boehm & W. Hahn & D. Root, Addgene 81779[89]), PEX26 (a gift from D. Sabatini, Addgene 139054[90]) and PACT (a gift from C. Norden, Addgene 105954[91]). Rabbit 2C1(1-27) and rat Cb5(100-134) were synthesized as G-blocks (IDT), while human eNOS(1-32) and Lck were introduced using primers and inserted into pRETROX TRE3G by Gibson assembly. NLRP3 mutants were subcloned into pcDNA3 using BamHI/EcoRI or BamHI/NotI sites for following ASC specks in the HEK293T system.

For the preparation of iBMDMs stably expressing fluorescently labeled ASC, ASC-mCerulean was amplified with PCR and subcloned into pLY093-EFS-BCMA-Blast-WPRE (pLenti Blast, a gift from Sidi Chen, Addgene 192194[92]) using BamHI/NheI sites to remove BCMA.

NLRP3 with forced oligomerization domains was generated by preparing a nucleotide sequence encoding a C-terminal fusion of NLRP3-YFP with TDP-43 (a gift from A. Gitler, Addgene 84912[69]) and inserted into pRETROX TRE3G by Gibson assembly.

R258W mutation within the NLRP3 was introduced with the site-directed mutagenesis. Point mutations K127A, K128A, K129A and K130A in NLRP3 were introduced by the site-directed mutagenesis, yielding 4xA mutant.

The truncated version of eNOS-NLRP3(1-686)-YFP (eNOS-(1-686)) was prepared using PCR and assembled into pRETROX TRE3G using Gibson assembly, whereas eNOS-NLRP3$^{F785A/D786R/F810A}$-YFP (F788/D789/ F813 in human) (eNOS-Δcage) and Walker A mutants NLRP3$^{K228A}$ and NLRP3$^{G227A/K228A/T229A}$ (NLRP3$^{3xA}$) were prepared with the site-directed mutagenesis with repliQa HiFi ToughMix (Quantabio, 732-3662).

All constructs were verified with Sanger sequencing (GATC/ Eurofins or Macrogen). Oligonucleotides used in this study are listed in Supplementary Table 3.

### Cell lines, retroviral and lentiviral transduction

Human kidney epithelial cells (HEK239T) (ATCC; CRL-11268) and iBMDM from female *Nlrp3*$^{-/-}$ mice (kind gift from K. A. Fitzgerald, University of Massachusetts Medical School, Worcester, MA, USA)[49] were cultured in DMEM supplemented with 10% FBS (Gibco) and maintained in a humidified incubator at 37 °C in 5% CO$_2$. All cell lines were routinely tested for mycoplasma contamination.

Doxycycline inducible NLRP3 mutants in NLRP3-KO iBMDMs were prepared with retroviral or lentiviral transduction[18]. Retroviral particles were prepared upon transfection of the Platinum-GP (Cell Biolabs) or Gryphon Ampho (Alelle Biotech) cells using 2.5 μg pRETROX TRE3G encoding NLRP3 mutants and 1.5 μg pCMV-VSV-G (Cell Biolabs) or 4 μg pRETROX TRE3G encoding NLRP3 mutants, respectively. NLRP3-KO iBMDMs expressing Tet-On 3G were seeded 4 × 10⁵ cells/well of a 6-well plate (Costar) and transduced the next day. The cells were further incubated for three days, when transduced cells were selected by growth in puromycin (6 μg/ml, Invivogen) and G418 (1.5 mg/ml, Roche) in DMEM supplemented with 10% FBS.

Lentiviral particles were prepared by transfecting Lenti-X 293 T (Takara-Bio, 632180) cells with 1.3 μg pLenti Blast, expressing ASC-mCerulean, 1.1 μg psPAX2 (a gift from Didier Trono, Addgene, 12260) and 0.4 μg pMD2.G (a gift from Didier Trono, Addgene 12259) using PEI MAX (Polysciences). After 48 h, lentiviruses were harvested, filtered through a 45-μm filter and added to iBMDMs expressing selected NLRP3 organelle-tethered variants that were seeded 4 × 105 cells/well of 6-well plate the day before. After three days, cells were selected by

growth in puromycin (6 μg/ml, Invivogen), blasticidin (6 μg/ml, Invivogen) and G418 (1.5 mg/ml, Roche) in DMEM supplemented with 10% FBS.

## Primary BMDMs

The laboratory for in vivo experiments at NIC is approved by the Administration of the Republic of Slovenia for Food Safety, Veterinary Sector and Plant Protection (Permit no. U34401-27/2013/13 and U34401-27/2013/19) as a breeding facility and facility for conducting experiments on laboratory mice. Laboratory mice were housed in the SPF facility in IVC cages GM500 (Techniplast) on a standard diet (Mucedola) and tap water was provided ad libitum. Mice were maintained in a 12–12 h dark-light cycle at ambient temperature (22 °C) and 40–60% relative humidity. Health/microbiological status was confirmed by Mouse Vivum immunocompetent panel (QM Diagnostics) as recommended by FELASA. Tissue collection for isolation of bone marrow was performed according to the permit for working with animal tissue and cells (U34401-11/2023/3). For isolation of bone marrow, two in-house bred 12-week-old female C57BL/6J OlaHSd mice were humanely euthanized by cervical dislocation. Bone marrow was flushed from the femurs and tibias. Bone marrow cells were cultured for 6 days in RPMI 1640 medium supplemented with 20% FBS, 40 ng/ml M-CSF (eBioscience), and penicillin/streptomycin (Gibco). Afterwards, BMDMs were either frozen or directly stimulated with LPS to follow the expression of endogenous NLRP3[18].

## Cell stimulation and NLRP3 inflammasome activation

Cells were plated at $1.5 \times 10^5$ cells per well in the 96-well plate (TPP) in the morning. The following evening, the cells were primed with ultra-pure LPS (100 ng/ml, Invivogen, tlrl-3pelps) in combination with doxycycline (0.5 or 1 μg/ml, Sigma-Aldrich, D9891-1G) in serum-free DMEM for 12–14 hours. After priming, the medium was exchanged with fresh DMEM containing the 10 μM nigericin (Sigma-Aldrich, N-7143) for 1 h or 200 μg/ml silica (Invivogen, tlrl-sio) or 20 μg/ml imiquimod (Invivogen, tlrl-imqs) for 6 h. After stimulation, the supernatants were collected for the ELISA and LDH assay.

## The effect of trafficking inhibitors

NLRP3-KO iBMDMs were plated and primed with LPS (100 ng/ml) and doxycycline (1 μg/ml) for 12 h, after which the medium was exchanged for fresh serum-free DMEM containing trafficking and NLRP3 inhibitors. After 4 h, nigericin was added (final concentration 10 μM) for 1 h, after which the medium was collected for assessment of IL-1β concentration. For the determination of the effect of trafficking compounds on pro-IL-1β expression, the experiment was performed using WT NLRP3 expressing NLRP3-KO iBMDMs in 24 well plates (TPP). After compound treatment, cells were lysed and the level of pro-IL-1β was followed by western blot.

The following compounds with final concentrations based on the previously published studies were used: golgicide A (20 μM, MedChemExpress, HY-100540) and AG1478 (20 μM, MedChemExpress, HY-13524) that act on Arf-GEF GBF1, brefeldin A (20 μM, MedChemExpress, HY-16592) that abolishes interaction of several ARF GTPase-GEF pairs, endocytosis-affecting compounds tyrphostin A23 (80 μM, MedChemExpress, HY-15644), dynasore (40 μM, Sigma-Aldrich, D7693) and pitstop2 (10 μM, MedChemExpress, HY-115604), and Retro-2 (20 μM, MedChemExpress, HY-122571) that affects early endosome to TGN trafficking, Vacuolin-1 (20 μM, MedChemExpress, HY-118630), NLRP3 inhibitor MCC950 (5 μM, Avistron, AV02509), deubiquitinase inhibitor G5 (10 μM, MedChemExpress, HY-100738).

## The effect of KCl and ROS inhibitors

NLRP3-KO iBMDMs were plated and primed with LPS (100 ng/ml) and doxycycline (1 μg/ml) for 11 h, after which the medium was exchanged for fresh serum-free DMEM containing KCl (30 and 50 mM final

concentration) and ROS inhibitors (100 μM PDTC, MedChemExpress, HY-18738, and 20 μM ebselen, Enzo, ALX-270-097-M025). After 30 min, activators nigericin (10 μM) or imiquimod (180 μM) were added. After 1 h, the supernatant was removed and IL-1β concentration was determined by ELISA.

## The effect of NLRP3 inhibitors on constitutive inflammasome activation

NLRP3-specific inhibitors were added 30 minutes before the addition of ultra-pure LPS (100 ng/ml) and doxycycline (1 μg/ml) in serum-free DMEM. After 11 hours, the secretion of IL-1β was followed by ELISA.

The following concentrations of inhibitors were used: MCC950 (10, 2, 0.5, 0.1 μM, Avistron, AV02509), Oridonin (2.5, 1.25, 0.5 μM, MedChemExpress, HY-N0004), RRx-001 (0.5, 0.25, 0.1 μM, MedChemExpress, HY-16438), CY-09 (6.25, 3.125, 1.56 μM, MedChemExpress, HY-103666), G5 (1.25, 0.63, 0.31 μM, MedChemExpress, HY-100738).

## Detection of cytokines by ELISA

Collected cell supernatants were used to measure concentrations of secreted cytokines with ELISA according to the manufacturer's instructions (IL-1β: Invitrogen, 88-7013-88, R&D Systems, DY401 and Biolegend, 432604; TNFα, IL-6 and IL-18: Invitrogen, 88-7324-88, 88-7064-88 and 88-50618-88, respectively).

## Detection of pyroptotic cell death by lactate dehydrogenase assay

Pyroptosis was assessed by measuring lactate dehydrogenase release using the CyQuant LDH cytotoxicity assay kit from Thermo Fisher (C20301), following the manufacturer's instructions. Simultaneously with nigericin treatment, cells were lysed by adding 10x lysis buffer as a positive control (dead cells). After 1 h, 50 μl of cell culture supernatant per well was transferred into a fresh 96-well plate (Thermo Fisher) and 50 μl of LDH assay buffer was added. Following incubation for 5–20 min, the reaction was stopped by the addition of 50 μl of stop solution to each well. Absorbance at 492 nm and 680 nm was measured using the BioTek Synergy Mx plate reader. The signal was normalized to LPS-treated cells as non-treated cells and lysed cells as dead cells.

## Detection of cell membrane perforation by propidium iodide uptake

Cells were plated into a black 96-well plate (Corning) at $1.5 \times 10^5$ cells per well. Following stimulation, propidium iodide (1 mg/ml, Thermo, P3566) was added to cells to achieve a final dilution of 1:300. After 30 minutes, cells were centrifuged for 5 min at $500 \times g$ and fluorescence was measured on Biotek Synergy Mx microtiter plate reader. The signal was normalized to LPS-treated cells as the non-treated control and lysed cells as positive control.

## Reconstitution of NLRP3 inflammasome in HEK293T cells and detection of ASC specks

The plasmid encoding ASC-mCerulean was a kind gift from E. Latz (Institute of Innate immunity, University of Bonn, Germany). $6 \times 10^5$ HEK293T cells were seeded onto poly-L-lysine (Sigma-Aldrich, P4707) coated μ-Slide 8 well (Ibidi) and transfected by polyethyleneimine (Sigma-Aldrich) or Lipofectamine 2000 (Invitrogen) with tethered NLRP3-YFP mutant (30 ng/well) and ASC-mCerulean (10 ng/well). After 16 h, 10 μM nigericin was carefully added to the medium. Cells were visualized with Leica TCS SP5 or Leica TSC SP8 (Leica Microsystems, Germany) within one hour post-treatment.

## Immunodetection of ASC specks in iBMDMs

$3 \times 10^5$ NLRP3-KO iBMDMs expressing NLRP3 mutants were seeded into poly-L-lysine-coated μ-Slide 8 well and stimulated with ultra-pure

LPS (100 ng/ml) and doxycycline (1 µg/ml) in serum-free DMEM. After 12 h, medium was exchanged for stimulation buffer (147 mM NaCl, 10 mM HEPES, 13 mM D-glucose, 2 mM KCl, 2 mM CaCl$_2$, 1 mM MgCl$_2$, pH 7.45[93]) with or without 10 µM nigericin. Cells were fixed with 4% paraformaldehyde (Electron Microscopy Sciences, 15714-S) for 15 min and permeabilized with 0.2% saponin and 1% BSA in PBS for 30 min. Cells were incubated with primary antibody against ASC (Anti-ASC Antibody, clone 2EI-7, Merck, 04-147, 1:200) for 2 h at room temperature, and secondary antibody (Alexa Fluor 488 goat anti-mouse IgG (H + L), Invitrogen, A11001, 1:200) for additional 2 h. Samples were further incubated with DAPI (300 nM) for 5 min and mounted with ibidi Mounting Medium. Images were acquired using LAS AF on Leica TCS SP8 (Leica Microsystems, Germany) with a Z step size of a 0.4 µm. ASC specks were manually counted in 5 random frames using Fiji v. 1.54p (NIH) software.

## Western blot

To detect the expression of NLRP3 variants, $7.5 \times 10^5$ or $3 \times 10^6$ cells per well were plated in 24-well or 6-well plates, respectively, and stimulated with ultra-pure LPS (100 ng/ml) in combination with doxycycline (1 µg/ml) in serum-free DMEM. After 12–14 h, cells were lysed and the protein concentration within the lysate was determined with the BCA method.

To detect cleaved caspase-1 and IL-1β, the cell supernatants were concentrated using 10k Amicon Ultra centrifugal filters (Millipore, Merck).

Protein samples (50 µg) were separated by SDS-PAGE, transferred onto nitrocellulose membrane (GE Healthcare), and incubated overnight with specific primary antibodies: NLRP3 (Cryo-2, Adipogen, AG-20B-0014-C100, 1:1000), caspase-1 p20 (Casper-1, Adipogen, AG-20B-0042-C100, 1:1000), ASC (Adipogen, AG-25B-0006-C100, 1:1000), IL-1β (GeneTex, 1G-GTX74034-100, 1:2000), β-Actin (Cell Signaling Technology, 8H10D10, 1:5000) and α/β-Tubulin (Cell Signaling Technology, 2148, 1:4000). The membranes were subsequently incubated for 1 h with the following HRP-conjugated secondary antibodies: anti-rabbit (Jackson ImmunoResearch, 111-035-003, 1:3000) and anti-mouse (Jackson ImmunoResearch, 115-035-003, 1:3000). HRP-labeled bands were detected with SuperSignal West Pico or SuperSignal West Femto Chemiluminescent Substrate (Thermo Scientific) using G-box (Syngene) and Genesnap 7.09 software. Uncropped immunoblots are shown in the Source Data file.

## Native PAGE

To determine the oligomerization state of NLRP3 variants, cells were seeded into a 6-well plate and primed overnight with LPS and doxycycline as described. All further steps were performed on ice. Cells were lysed in 0.1 % CHAPS buffer (0.1% CHAPS, 20 mM HEPES-KOH, 5 mM MgCl$_2$, 0.5 mM EGTA) with added protease inhibitors (Roche cOmplete, Mini Protease Inhibitor Cocktail, 04693124001) and syringed at least 10-times through G26 needle. After the addition of native PAGE loading dye, samples were separated on a 6% separating gel for 5–6 h at 90 V on ice. After native PAGE, samples were blotted onto a nitrocellulose membrane and detected using anti-NLRP3 primary antibodies.

## Confocal microscopy

To follow the localization of the NLRP3 variants, $3 \times 10^5$ NLRP3-KO iBMDMs expressing NLRP3 mutants were seeded into µ-Slide 8-well. Cells were treated with ultra-pure LPS (100 ng/ml) and doxycycline (1 µg/ml) overnight. The following morning, cells were incubated with organelle-specific dyes for 30–40 min followed by a series of washes and lastly stained with Hoechst 33342 (1 µg/ml; Immunochemistry Technologies, 639) for 5 min with or without propidium iodide (1 µg/ml) for 30 min. For the endoplasmic reticulum, cells were incubated with ER-Tracker Red (Invitrogen, E34250, 1:2000). Golgi apparatus was stained with red fluorescent BODIPY TR C$_5$ (Invitrogen, B-34400, 1:100). Plasma membrane was observed with Cholera Toxin Subunit B, Alexa Fluor 647 (Invitrogen, C34778, 1:200). Lysosomes were stained with LysoTracker Deep Red (Invitrogen, L12492, 1:20,000) and mitochondria with MitoTracker Deep Red (Invitrogen, M22426, 1:2000). Tubulin Tracker Deep Red (Invitrogen, T34076, 1:1000) was used to determine location of MTOC, seen as γ-tubulin foci. After staining, cells were incubated with canonical activators nigericin (10 µM) or imiquimod (20 µg/ml) and imaged within 1 h or 1.5 h post-addition, respectively. Colocalization of peroxisomes was determined with transient transfection of HEK293T cells using mCerulean-Peroxisomes-2 (a gift from M. Davidson, Addgene 54691) and plasmids encoding WT or peroxisome-enriched NLRP3 variants. Cells were observed using Leica TCS SP5 or Leica TSC SP8 confocal microscopes (Leica Microsystems, Germany). Image processing was performed in LAS X v. 3.7.4 (Leica Microsystems) and Fiji v. 1.54p (NIH). The percentage of NLRP3 puncta was determined manually in 5 random frames, where the number of cells with puncta was divided by the number of cells expressing NLRP3.

## Flow cytometry

Cells ($0.6 \times 10^6$ per well of a 24-well microtiter plate) were seeded and primed overnight with LPS (100 ng/ml) and doxycycline (1 µg/ml). The next day, cells were detached in PBS/2.5 mM EDTA and YFP expression was followed using Cytek Aurora (Cytek Biosciences). Data was acquired and unmixed using SpectroFlo v3.1.0 software (Cytek Biosciences). The resulting unmixed (empty vector cells with the addition of doxycycline were used as negative control for unmixing) and raw FCS files were analyzed using manual gating in FlowJo v. 10.10.0 software (BD Biosciences) according to the gating strategy presented in Supplementary Fig. 6a. After the selection of the population representing macrophages (debris was removed), doublets were excluded, and singlet population was analyzed for YFP fluorescence.

## Transmission electron microscopy (TEM)

NLRP3-KO iBMDMs expressing NLRP3 mutants were seeded into tissue culture dishes with a 40 mm diameter (TPP) at a seeding density of 4 ×10$^5$ cells per culture dish. Cells were stimulated with ultra-pure LPS (100 ng/ml) and doxycycline (1 µg/ml) in serum-free DMEM overnight. Nigericin was added the next morning to a final concentration of 10 µM. After 45 minutes, the cells were rinsed with culture medium and fixed with 4.5% (w/v) formaldehyde and 2% (v/v) glutaraldehyde in 0.1 M cacodylate buffer (pH 7.4) for 3 h at 4 °C. The fixation was followed by overnight rinsing in 0.33 M sucrose in 0.1 M cacodylate buffer at 4 °C. The samples were then post-fixed in 1% (w/v) osmium tetroxide for 1 h at 4 °C, rinsed in distilled water for a few seconds, incubated in 2% uranyl acetate for 1 h at room temperature and rinsed again with distilled water for few seconds. Next, the samples were dehydrated in a graded ethanol series (50, 70, 90%) for 15 minutes each and 100% ethanol for 30 minutes, 2 times. Dehydrated samples were embedded in Epon (Serva Electrophoresis, Heidelberg, Germany) by infiltration (immersion in Epon for 10 min, repeated 3 times, followed by a 30-min immersion at room temperature). The polymerization of Epon was performed over the next 5 days with gradual temperature increase (35 °C, 45 °C, 60 °C, 70 °C and 80 °C) every 24 h. Afterward, the 60 nm ultrathin sections were prepared with Ultramicrotome (Leica EM UC6), contrasted with uranyl acetate and lead citrate and examined at the operation voltage 80 kV with a Philips CM100 electron microscope (Philips, Eindhoven, The Netherlands) equipped with the CCD camera (AMT, Danvers, MA, United States).

## Statistical analysis

Data are presented as mean ± SEM of different experiments, where a single point represents the average of technical replicates performed within the single experiment. Statistical tests were calculated with GraphPad Prism (GraphPad Software, version 8.4.3). Statistical

significance was examined using an unpaired two-tailed *t* test. In cases where variances differed between groups, an unpaired two-tailed *t* test with Welch correction was employed. Statistical details for each experiment can be found in the figure legends and Source data.

**Reporting summary**

Further information on research design is available in the Nature Portfolio Reporting Summary linked to this article.

## Data availability

All data are included in the Supplementary Information or available from the authors, as are unique reagents used in this Article. The raw numbers for charts and graphs are available in the Source Data file whenever possible. The data generated in this study are provided in the manuscript, Supplementary Information and in the Source Data file. Technical replicates from independent biological repeats have been included in the Source Data for all experiments in the main figures. All unique materials are available from the corresponding author upon request. Source data are provided with this paper.

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

## Acknowledgements

The authors thank the members of the Department of Synthetic Biology and Immunology, Prof. T. Bratkovič (University of Ljubljana, Faculty of Pharmacy, Slovenia) and Assist. Prof. N. Kraševec (National Institute of Chemistry, Slovenia) for valuable discussions. The authors thank B. Anderle (née Alić) and K. Glavič for the preparation of some of the constructs in the scope of their MSc theses. The authors are grateful to R. Bremšak, I. Škraba, T. Strmljan, T. Kos, A. Babnik, A. Maučec and G. Junger (National Institute of Chemistry, Slovenia) and N. Pavlica Dubarić, M. Radanović, S. Železnik and S. Čabraja (Institute of Cell Biology, Faculty of Medicine, University of Ljubljana) for technical assistance. The authors would like to thank Prof. K. A. Fitzgerald (University of Massachusetts Medical School, USA) for providing immortalized mouse macrophages, Prof. E. Latz (Institute of Innate Immunity, University of Bonn, Germany) for the plasmid encoding ASC-mCerulean, and T. Fink (National Institute of Chemistry, Slovenia) for the help with lentiviral transduction. This work was funded by: Slovenian Research and Innovation Agency (ARIS) project grant J3-1746 (I.H.B.), ARIS project grant N3-0358 (I.H.B.), ARIS project grant J3-60056 (I.H.B.), ARIS young researcher's PhD grants (E.B., S.O.), ARIS project grant Z3-4501 (T.Ž.R.), ARIS research core funding P3-0108 (M.E.K.), ARIS research core funding P4-0176 (R.J.), EU HORIZON-WIDERA CTGCT project 101059842 (R.J.), the European Research Council (ERC) Advanced Grant NalpACT (M.G.) and the Deutsche Forschungsgemeinschaft (DFG) under Germany's Excellence Strategy–EXC2151–390873048 (M.G.). I.H.B. and E.B. are members of COST PRESTO Action CA21130. I.H.B. is grateful to the European Federation of Immunological Societies for the Eastern Star Award 2022.

## Author contributions

Conceptualization: I.H.B.; Methodology: E.B., M.E.K., I.H.B.; Investigation: E.B., T.Ž.R., S.O., M.E.K., I.H.B.; Visualization: E.B., M.E.K., I.H.B.; Funding acquisition: T.Ž.R., M.E.K., M.G., R.J., I.H.B.; Project administration: I.H.B.; Supervision: I.H.B.; Writing-original draft: E.B., I.H.B.; Writing-review & editing: E.B., T.Ž.R., S.O., M.E.K., M.G., R.J., I.H.B.

## Competing interests

M.G. is a co-founder of IFM Therapeutics and an advisor to BioAge Labs. The other authors declare no competing interests.
