## [Transparent Peer Review file · Nature Communications]

Clustering of NLRP3 induced by membrane or protein scaffolds promotes inflammasome assembly

Corresponding Author: Dr Iva Hafner Bratkovič

Version 0:

Reviewer comments:

Reviewer #1

(Remarks to the Author)

Borsic and colleagues address an important question in the inflammasome field. The authors demonstrate that restriction of NLRP3 to a variety of membranes including mitochondria, ER, GA, lysosomes, peroxisomes, and plasma membrane still allows for efficient inflammasome activation. This suggests that a precise subcellular location for NLRP3 is not required for its assembly and activation.

The manuscript is well written and the logic easy to follow. The data are clear and the interpretations appropriate. I have a few minor comments:

- 1) The authors show LDH release as normalized to WT. This does not allow the reader to determine how much cell death is occurring. It would be better to show this as % cell death.
- 2) Figure 1 legend should read "The mean +/- SEM of three independent experiments is shown (D-I)."
- 3) The authors should comment on the location of the various membrane targeted NLRP3 variants after LPS-priming and activation with the second signal (nigericin, silica, imiquimod). Do they remain at their targeted membrane?
- 4) What is the subcellular location of ASC speck formation following activation in cells expressing the NLRP3 variants? Is there a single ASC speck or do multiple ASC specks form?

Reviewer #2

(Remarks to the Author)

This manuscript investigates the role of subcellular location on influencing NLRP3 activation. This research is of interest because despite the importance of NLRP3 in worsening disease, and the enduring progress in the field to understanding how NLRP3 is regulated, it is still very unclear what exactly the subcellular events are that dictate activation NLRP3 inflammasome. As highlighted by the authors, multiple distinct organelles have been implicated as important for inflammasome activation, and recent studies are suggesting NLRP3 recruitment to organelle membranes could be a critical step in inflammasome activation. Therefore, Boršič et al present a very timely study to determine the impact of targeting NLRP3 to diverse organelles on inflammasome activation. Overall, this study has been executed well with appropriate controls, the data is convincing, sufficiently described and provides an advancement in our understanding of NLRP3 activation. Of particular strength is the doxycycline-inducible NLRP3-YFP constructs allowing controlled, low level NLRP3 expression, minimising the risk of over-expression artefacts and allowing the detection of constructs which cause spontaneous NLRP3 activation. I have a couple of minor points which the authors should consider to improve their manuscript.

1. Whilst Boršič et al demonstrate that their constructs work as expected to target NLRP3 to organelles of interest, these experiments were only performed in LPS-primed macrophages, but not following NLRP3-activating stimuli (e.g. nigericin, imiquimod). Therefore, it is currently not possible to determine if the organelle-targeted localisation is maintained following stimulation, or if the activating stimuli is sufficient to override the organelle-targeting and cause NLRP3 re-localisation to a shared site to the WT. This is an important point as may determine if organelle targeting is a passive event (i.e. doesn't matter where NLRP3 is, as targeting can be overridden by activating stimuli), or if localisation at organelle membranes is a pre-requisite step (i.e. NLRP3 must be on any membrane, so stays where it is targeted). It is suggested that the authors test

localisation of NLRP3-YFP in their organelle-targeted macrophages following NLRP3 stimulation to see if the inflammasome forms at the site of the respective organelle. Further, it would be very interesting to see if the position of the ASC speck correlated to the site of the targeted organelle.

2. A disadvantage is the varying expression levels of NLRP3 in their system resulting in varied IL-1 β release and LDH. It is unclear if the varying expression is due to fewer cells expressing the construct, or if less NLRP3 is expressed per cell - Have these cell lines undergone monoclonal selection? The difference in NLRP3 expression may make some more subtle changes induced by organelle targeting harder to identify. Ideally these should be expressed at the same level, but in the absence of this, is there anyway the authors can correlate the level of NLRP3 expression with IL-1b release etc to provide more confidence that inflammasome activation is unchanged with organelle targeting (aside from the Golgb1/PACT/TPD54 constructs)?

3. Some datasets are only of 2 individual repeats – it is encouraged that all datasets should be reflected of at least 3 independent experiments.

Reviewer #3

(Remarks to the Author)

In the manuscript “Clustering of NLRP3 on diverse cellular scaffolds is a key step toward inflammasome formation”, using murine NLRP3-deficient iBMDMs reconstituted with modified versions of NLRP3, Boršić et al. study the activation process of the NLRP3 inflammasome. They show NLRP3 inflammasome activation can occur independently of its subcellular localization and mode of activation. Forced association of NLRP3 with membranes is not sufficient for activation, but NLRP3 activation becomes independent of the positively charged cluster in the linker region when scaffold clustering is facilitated by other means. They further show that NLRP3 activation does not only work on a membranous scaffold but could also occur with a protein-based scaffold. Using established NLRP3 inhibitors binding the NACHT domain, they demonstrate that clustering of NLRP3 destabilizes intramolecular interactions keeping the NACHT domain in inactive conformation. They propose a mechanism of activation where cellular disbalance promotes NLRP3 clustering on a scaffold, leading to rearrangements of NLRP3 and subsequent activation. The experiments are well-conducted and support the conclusions. This is an important study that brings the field forward and I believe will be well-received. The central limitation is that the clustering methods are essentially artificial and it is difficult to ascertain the precise biological relevance. Specifically, the question remains open if, under physiological conditions, there is only one physiological scaffold for NLRP3 clustering, or if NLRP3 activation can be supported by scaffolding signals from different compartments. Furthermore, the question which (if any) danger information is conveyed to / sensed by NLRP3 through scaffolding remains open. However, these questions are open in the field for close to 20 years now and this study is an important step towards answering them.

Major points:

1. The authors are understandably mostly careful in their interpretation whether clustering of NLRP3 on a scaffold is a part of licensing (required to get NLRP3 into a state ready to sense the activating signal but not itself activating it) or if scaffold clustering is part of activation itself. The data presented together with hints from previous publications strongly suggests scaffold clustering is required but not sufficient. Connecting the present study to a very recent publication (PMID 39571574) might help shed light on this question. Saller et al. suggest a two-activation-signal model of NLRP3 activation in addition to and distinct from the established priming and activation model. They propose that a mitochondrial signal (they call signal 2a) and an additional signal (2b) are sufficient for NLRP3 activation. The authors could try to combine their anchor-tagged versions of NLRP3 with OXPHOS inhibitors such as antimycin or oligomycin, or with resiquimod, and monitor inflammasome activation. If a combination of anchor-tagged NLRP3 with either OXPHOS inhibitors or with resiquimod would be sufficient for activation, this would allow an interpretation of the signals involved in the actual activation of NLRP3.

2. The data presented for cytokines and LDH is most of the time presented as normalized to a control condition. I don't see the advantage and would prefer to see the actual values. Could the authors provide unnormalized data in the supplement, or at least to the reviewer? Is there a specific reason why some panels were not normalized but most were (i.e., Fig. 3A as compared to 3C)?

3. Some experiments were only performed twice (n=2). This is understandable for very complex or expensive experiments, but at least for the main figure panels, it would be preferable to have performed them at least three times (with consistent results).

4. In the experiments using KCl (Fig. 4C-H), I could not find the final extracellular KCl concentration used. Please specify in the figure, legend or methods section (final concentration, considering 5 mM in media). Extracellular KCl experiments are prone to false-positive results, i.e., it interfering with general cellular function. Controls that the KCl treatment does not interfere with other cellular functions (e.g., Aim2 or TLR signaling), should be included. The KCl data presented in this study is, however, not essential to the interpretation and conclusions and could also be omitted without reducing the impact of the study. Note that silica and other particulate/crystalline activators were reported as a KCl efflux-dependent NLRP3 activator, while from the text, one gets the impression these were independent (line 315-317).

Minor points:

5. In the summary, please check the sentence “We further reveal the crucial membrane or protein scaffold-mediated clustering that leads to the unfastening of the inactive NACHT domain conformation.” The meaning is unclear to me. Does it refer to the next sentence? Or is this a teaser and the reader should look for it in the main text? Or does the sentence stand alone? If stand alone, please check the wording since the message is not clear. Maybe “We further reveal that membrane or protein scaffold-mediated clustering is crucial since it leads to the unfastening of the inactive NACHT domain conformation.”? Please also check the last sentence of the summary and maybe replace “facilitates”.

6. Is activation on anchored NLRP3 still ROS dependent, i.e., sensitive to inhibitors such as PDTC or ebselen?

Version 1:

Reviewer comments:

Reviewer #1

(Remarks to the Author)

The authors have addressed all of my previous concerns.

Reviewer #2

(Remarks to the Author)

Boršić et al have addressed all of my major concerns and have produced a very exciting manuscript that demonstrates new interesting findings to the community.

I only have a couple of minor image quality/processing concerns that should be addressed:

1) The addition of immunofluorescence to show localisation of NLRP3 following activation improves the manuscript and gives confidence that organelle targeting is maintained. Whilst the data is encouraging, the representative images in the main figure are hard to interpret in places due to quality of the images. If higher quality images are not available, I suggest that separate stains are shown to make this data easier to interpret, similar as seen for example in the supplementary figure 2. If not possible due to space restrictions, addition of separated images as an additional supplementary figure would be acceptable.

2) Likewise, supplementary figure 11 shows exciting data suggesting ASC speck positioning is also localised to organelle targeting, but this is hard to see due to the size of the images. Perhaps this figure can be split up to ensure that the images can be seen clearly.

3) ASC speck staining in supplementary figure 9 is dim – can the ASC stain be brightened in this figure.

Reviewer #3

(Remarks to the Author)

The authors answered all my questions satisfactorily.

Dear reviewers,

We would like to thank you for the constructive comments that helped us to improve the study. As you will see, we addressed a large majority of your concerns experimentally and incorporated new results in the revised version of the manuscript. Given that we were already close to the recommended word limit, we tried to be as succinct as possible when describing the newly acquired results, while not losing on the interpretation of the results. To include new results in the main figures, some previous results were transferred to the Supplementary information, which has now been extended.

Reviewers 1 and 2 were particularly interested in whether engineered NLRP3 remains localized to targeted organelles after inflammasome stimulation, and how this spatial restriction of NLRP3 affects ASC speck formation. We now show that, upon treatment with canonical triggers, organelle-tethered NLRP3 variants are retained at targeted membranes, despite changes in organellar structure. Furthermore, ASC specks are formed in the vicinity, proving that organelle-tethered NLRP3 variants assemble the inflammasome on the designated membranes. These microscopic images have been added to Figs. 1 and 2 and Suppl. Figs. 2 and 5. We updated representative images for the WT and lysosome-tethered variants in Figure 2c due to a change in the dye used for lysosomal staining. We have also addressed the differences in expression levels of our NLRP3 variants as raised by reviewer 2 (Suppl. Fig. 6).

Addressing reviewer 3 comment, we show that WT and localized NLRP3 variants are still dependent on K⁺ efflux for nigericin activation, suggesting a shared activation mechanism (Fig. 3f, Suppl. Fig. 13). We also demonstrated that IL-1 β release induced by either nigericin or imiquimod treatment is suppressed by ROS inhibitors in both WT and tethered NLRP3 variants (Fig. 3f, Suppl. Fig. 13).

While our manuscript was in review, we also worked on obtaining further evidence that activation of NLRP3 with forced clustering is not an artifact of tag-induced oligomerization. To test this, we prepared Walker A mutants, where NLRP3's ability to bind ATP is disrupted. These variants lost their constitutive activity, which could not be rescued by nigericin. This demonstrates that as in trigger-induced inflammasome activation, forced clustering-induced activation also required ATP binding. We included these data in Figure 6k.

We have added the previously missing biological repeats (some also incorporated into the Source data file for demonstration) as requested by reviewers 2 and 3, and included non-normalized data, as requested by reviewers 1 and 3.

Below, we provide a detailed point-by-point response to each of your comments. We have marked the new and revised text in the manuscript file in red. As some minor changes (such as article and figure changes and deleted text) are not visible in this file, we also provide a tracked changes version of the manuscript as a supplementary file.

REVIEWER COMMENTS

Reviewer #1 (Remarks to the Author):

Borsic and colleagues address an important question in the inflammasome field. The authors demonstrate that restriction of NLRP3 to a variety of membranes including mitochondria, ER, GA, lysosomes, peroxisomes, and plasma membrane still allows for efficient inflammasome activation. This

suggests that a precise subcellular location for NLRP3 is not required for its assembly and activation.

The manuscript is well written and the logic easy to follow. The data are clear and the interpretations appropriate. I have a few minor comments:

Authors' response: We sincerely thank the reviewer for recognizing the importance of our study and for the thoughtful, constructive feedback. We appreciate the kind and valuable comments, which helped us improve the manuscript.

R-1.1. The authors show LDH release as normalized to WT. This does not allow the reader to determine how much cell death is occurring. It would be better to show this as % cell death.

A-1.1. We used normalization to WT signal since we were comparing localization-restricted variants to non-restricted NLRP3 (WT) and this enabled us to reduce interexperimental variation (e.g. different LPS stocks, doxycycline stocks etc). However, we agree that some information is lost. We have now included important non-normalized data from Figures 1-3 in either supplementary file (for normalized figures shown in the main manuscript) or the source data (for normalized data shown in the supplement). Additionally, we have moved the LDH data to the supplement due to additional microscopic images that were added to figures 1 and 2. In general, the LDH% are rather low, but this is not surprising for nigericin stimulation in growth medium.

R-1.2. Figure 1 legend should read "The mean +/- SEM of three independent experiments is shown (D-I)."

A-1.2. We thank the reviewer for noticing this. The figure legend has been corrected and updated according to the revised figure.

R-1.3. The authors should comment on the location of the various membrane targeted NLRP3 variants after LPS-priming and activation with the second signal (nigericin, silica, imiquimod). Do they remain at their targeted membrane?

A-1.3. We followed the localization of membrane-targeted NLRP3-YFP after the addition of nigericin (Figures 1c-e; 2 c, d) and imiquimod (Supplementary Figures 2, 5). Despite the changes that are happening to some organelles, NLRP3-YFP signal localized with organelle markers, demonstrating that even after the second signal NLRP3 is still associated with membranes. We also tried to visualize cells after silica treatment, but the presence of silica particles prevented us from visualizing the cells without interference.

R-1.4. What is the subcellular location of ASC speck formation following activation in cells expressing the NLRP3 variants? Is there a single ASC speck or do multiple ASC specks form?

A-1.4. In order to follow ASC speck formation downstream of tethered NLRP3 activation in living cells, we transduced selected cell lines with ASC-mCerulean encoding lentivirus. As shown in Supplementary Fig. 11, fluorescent ASC specks are located proximally to the organelle membranes to which NLRP3 was targeted. To observe ASC speck formation of endogenous non-tagged ASC, selected cell lines were activated with nigericin and fixed after 20 minutes. The number of ASC speck containing cells was

comparable between the cell lines, further supporting the conclusion that spatially restricting NLRP3 does not hamper inflammasome activation. The majority of cells contained a single speck, some had more, but the observation was not specific to any cell line and was also present in the location non-restricted NLRP3 cell line.

Reviewer #2 (Remarks to the Author):

This manuscript investigates the role of subcellular location on influencing NLRP3 activation. This research is of interest because despite the importance of NLRP3 in worsening disease, and the enduring progress in the field to understanding how NLRP3 is regulated, it is still very unclear what exactly the subcellular events are that dictate activation NLRP3 inflammasome. As highlighted by the authors, multiple distinct organelles have been implicated as important for inflammasome activation, and recent studies are suggesting NLRP3 recruitment to organelle membranes could be a critical step in inflammasome activation. Therefore, Boršić et al present a very timely study to determine the impact of targeting NLRP3 to diverse organelles on inflammasome activation. Overall, this study has been executed well with appropriate controls, the data is convincing, sufficiently described and provides an advancement in our understanding of NLRP3 activation. Of particular strength is the doxycycline-inducible NLRP3-YFP constructs allowing controlled, low level NLRP3 expression, minimising the risk of over-expression artefacts and allowing the detection of constructs which cause spontaneous NLRP3 activation. I have a couple of minor points which the authors should consider to improve their manuscript.

Authors' response: We thank the reviewer for the positive and encouraging evaluation of our manuscript. We are grateful for the detailed and constructive comments, which helped us further strengthen the overall quality and coherence of our study.

R2-1. Whilst Boršić et al demonstrate that their constructs work as expected to target NLRP3 to organelles of interest, these experiments were only performed in LPS-primed macrophages, but not following NLRP3-activating stimuli (e.g. nigericin, imiquimod). Therefore, it is currently not possible to determine if the organelle-targeted localisation is maintained following stimulation, or if the activating stimuli is sufficient to override the organelle-targeting and cause NLRP3 re-localisation to a shared site to the WT. This is an important point as may determine if organelle targeting is a passive event (i.e. doesn't matter where NLRP3 is, as targeting can be overridden by activating stimuli), or if localisation at organelle membranes is a pre-requisite step (i.e. NLRP3 must be on any membrane, so stays where it is targeted). It is suggested that the authors test localisation of NLRP3-YFP in their organelle-targeted macrophages following NLRP3 stimulation to see if the inflammasome forms at the site of the respective organelle. Further, it would be very interesting to see if the position of the ASC speck correlated to the site of the targeted organelle.

A2-1. We thank the reviewer for this important comment. To determine whether NLRP3 location changed upon stimulation, we tested two different stimuli, nigericin and imiquimod. Despite structural changes happening within the cells, consistent with previously published and within the manuscript cited studies, tethered NLRP3 variants colocalized with organelle markers (Fig. 1c-e, 2c-d, Supplementary Fig. 2a-c, 5c-d), suggesting that NLRP3 variants remain tethered to membranes during

the activation process. Furthermore, using fluorescently labelled ASC, we were able to demonstrate that ASC specks are formed in the vicinity of NLRP3 residing organelle (Suppl. Fig. 11).

R-2.2. A disadvantage is the varying expression levels of NLRP3 in their system resulting in varied IL-1 β release and LDH. It is unclear if the varying expression is due to fewer cells expressing the construct, or if less NLRP3 is expressed per cell - Have these cell lines undergone monoclonal selection? The difference in NLRP3 expression may make some more subtle changes induced by organelle targeting harder to identify. Ideally these should be expressed at the same level, but in the absence of this, is there anyway the authors can correlate the level of NLRP3 expression with IL-1 β release etc to provide more confidence that inflammasome activation is unchanged with organelle targeting (aside from the Golgb1/PACT/TPD54 constructs)?

A-2.2. Within our study, we demonstrate that NLRP3, while restricted to specific organelles, can still support inflammasome activation. However, as differences between activation levels of different variants seem to be a characteristic of a particular variant more than connected to a particular organelle, we cannot state that a particular location is preferred over another.

As we demonstrate, the ability of variants to respond and assemble an inflammasome depends on many factors, such as the position of the tag (N-terminus, C-terminus), the linker and also the expression level of NLRP3 variants. Although we did not make single clones for each specific localized variant, we used two different single-cloned Tet3G transduced NLRP3-KO cell lines, and we transduced the series of NLRP3 variants at least three different times, obtaining similar responses. We believe our system is thus very robust and the results that show the average of the means of three independent experiments are representative. Additionally, similar trends in the level of activation were observed in NLRP3^{R258W} organelle mutants that do not require the second trigger, which is a good estimation of the effect of tagging on the ability of NLRP3 variant to undergo appropriate conformational change as well as downstream stages of inflammasome assembly (e.g. ASC and procaspase-1 recruitment).

As NLRP3 variants are tagged with YFP, we used flow cytometry aside from already presented western blots to provide additional information on the individual cell level. As expected from the retroviral transduction, where only a few copies of the gene are integrated into the genome, the majority of cell lines exhibit similar expression profiles, ranging from low to middle expression, while several have lower overall expression, such as Tmem192-tagged NLRP3 (Supplementary Figure 6). To make an additional comparison between WT and the lowest expressing variant Tmem192, we made several single clones, but unfortunately, we were not able to obtain single clones that would reach similar expression levels. Thus, we switched to testing different doxycycline concentrations and followed protein expression on western blot and simultaneously IL-1 β release after nigericin stimulation. Although Tmem192-tagged variant could not reach the same expression level as the WT, we observed a clear correlation between NLRP3 expression level and IL-1 β release.

R-2.3. Some datasets are only of 2 individual repeats-it is encouraged that all datasets should be reflected of at least 3 independent experiments.

A-2.3 We have performed additional repeats of experiments to ensure that the majority of datasets presented in the main figures are based on at least three independent experiments. For several experiments, such as the inflammasome activation followed by western blot (Fig. 3c) we included all blots from the three independent experiments in the source data.

Reviewer #3 (Remarks to the Author):

In the manuscript "Clustering of NLRP3 on diverse cellular scaffolds is a key step toward inflammasome formation", using murine NLRP3-deficient iBMDMs reconstituted with modified versions of NLRP3, Boršić et al. study the activation process of the NLRP3 inflammasome. They show NLRP3 inflammasome activation can occur independently of its subcellular localization and mode of activation. Forced association of NLRP3 with membranes is not sufficient for activation, but NLRP3 activation becomes independent of the positively charged cluster in the linker region when scaffold clustering is facilitated by other means. They further show that NLRP3 activation does not only work on a membranous scaffold but could also occur with a protein-based scaffold. Using established NLRP3 inhibitors binding the NACHT domain, they demonstrate that clustering of NLRP3 destabilizes intramolecular interactions keeping the NACHT domain in inactive conformation. They propose a mechanism of activation where cellular disbalance promotes NLRP3 clustering on a scaffold, leading to rearrangements of NLRP3 and subsequent activation. The experiments are well-conducted and support the conclusions.

This is an important study that brings the field forward and I believe will be well-received. The central limitation is that the clustering methods are essentially artificial and it is difficult to ascertain the precise biological relevance. Specifically, the question remains open if, under physiological conditions, there is only one physiological scaffold for NLRP3 clustering, or if NLRP3 activation can be supported by scaffolding signals from different compartments. Furthermore, the question which (if any) danger information is conveyed to / sensed by NLRP3 through scaffolding remains open. However, these questions are open in the field for close to 20 years now and this study is an important step towards answering them.

Authors' response: We are grateful to reviewer for the thoughtful and comprehensive evaluation of our manuscript as well as for emphasizing the importance of our study. The large majority of previous studies have been performed using endogenous NLRP3 which led to very different conclusions regarding the role of the specific organelle in inflammasome activation. Thus, we used a synthetic immunology approach to provide further insight into how NLRP3 location impacts inflammasome activation. We believe that the conclusions of our study consolidate previous seemingly contradicting studies reporting on the location and the contribution of specific organelles in NLRP3 inflammasome activation.

Major points:

R-3.1. The authors are understandably mostly careful in their interpretation whether clustering of NLRP3 on a scaffold is a part of licensing (required to get NLRP3 into a state ready to sense the activating signal but not itself activating it) or if scaffold clustering is part of activation itself. The data presented together with hints from previous publications strongly suggests scaffold clustering is required but not sufficient. Connecting the present study to a very recent publication (PMID 39571574) might help shed light on this question. Saller et al. suggest a two-activation-signal model of NLRP3 activation in addition to and distinct from the established priming and activation model. They propose that a mitochondrial signal (they call signal 2a) and an additional signal (2b) are sufficient for NLRP3 activation. The authors could try to combine their anchor-tagged versions of NLRP3 with OXPHOS inhibitors such as antimycin or oligomycin, or with resiquimod, and monitor inflammasome activation. If a combination of anchor-tagged NLRP3 with either OXPHOS inhibitors or with resiquimod would be

sufficient for activation, this would allow an interpretation of the signals involved in the actual activation of NLRP3.

A-3.1 As nicely summarized by the reviewer, other studies and the current manuscript suggest that binding of NLRP3 to various scaffolds is not sufficient for activation. Including the new data with Walker A mutants (Fig. 6k) we now provide additional evidence that clustering likely induces unfastening of the closed conformation but still requires the ATP binding for inflammasome activation. We were very eager to investigate the two-signal activation model, as suggested, as it could further clarify in which step of inflammasome activation the clustering is engaged. We were able to reproduce the study in primary BMDMs, which are usually more sensitive than immortalized cell lines. Using our membrane-tethered NLRP3 variant iBMDMs we tested a wide range of experimental conditions, including different priming, concentration of compounds and timing parameters, but we were not successful. Concretely, low IL-1 β levels released following treatment with a single agent or a combination of resiquimod and OXPLOS inhibitors (rotenone, antimycin A, oligomycin) prevented making any certain conclusions. As a control, we used imiquimod that potently induced inflammasome activation under all tested conditions, confirming the functionality of the experimental setup. We referenced this important study in the introduction and will try to get further insight in the relationship of NLRP3 location and the newly discovered two-signal activation model in our future work.

R-3.2. The data presented for cytokines and LDH is most of the time presented as normalized to a control condition. I don't see the advantage and would prefer to see the actual values. Could the authors provide unnormalized data in the supplement, or at least to the reviewer? Is there a specific reason why some panels were not normalized but most were (i.e., Fig. 3A as compared to 3C)?

A-3.2. As we are most of the time comparing the level of activation between non-localized (WT) and localization-restricted NLRP3 variants, we chose to normalize the data to WT within the same experiment. This also accounts for variations due to experimental conditions (e.g. different LPS or doxycycline stocks). However, we understand the importance of presenting the non-normalized data, so we have now included the majority of non-normalized data for figures 1-3 in either supplementary data or in the source data, while keeping the normalized data in the main figures. The trends in activation remain consistent regardless of the method used to present the results.

R-3.3. Some experiments were only performed twice (n=2). This is understandable for very complex or expensive experiments, but at least for the main figure panels, it would be preferable to have performed them at least three times (with consistent results).

A-3.3. We conducted additional repetitions of the experiments to ensure that for the majority of experiments the data represent either the means or a representative result from at least three independent experiments. For certain experiments, such as the inflammasome activation assay with western blot analysis (Fig. 3c), we have also included all blots from three independent experiments in the source data.

R-3.4. In the experiments using KCl (Fig. 4C-H), I could not find the final extracellular KCl concentration used. Please specify in the figure, legend or methods section (final concentration, considering 5 mM in media). Extracellular KCl experiments are prone to false-positive results, i.e., it interfering with general cellular function. Controls that the KCl treatment does not interfere with other cellular functions (e.g., Aim2 or TLR signaling), should be included. The KCl data presented in this study is,

however, not essential to the interpretation and conclusions and could also be omitted without reducing the impact of the study. Note that silica and other particulate/crystalline activators were reported as a KCl efflux-dependent NLRP3 activator, while from the text, one gets the impression these were independent (line 315-317).

A-3.4. We thank the reviewer for this comment. We conducted a new experiment using K⁺-independent NLRP3 trigger imiquimod to demonstrate that added K⁺ concentrations (30 mM and 50 mM) do not have unspecific effects on other processes, such as IL-1 β levels. This new data is now included in Figure 3e and Supplementary Figure 9f-k. As the sentence preceding the unclear statement was removed (since K⁺-efflux data are now shown in another figure), we believe that the text regarding particulate activators could not be misinterpreted.

Minor points:

R-3.5. In the summary, please check the sentence “We further reveal the crucial membrane or protein scaffold-mediated clustering that leads to the unfastening of the inactive NACHT domain conformation.” The meaning is unclear to me. Does it refer to the next sentence? Or is this a teaser and the reader should look for it in the main text? Or does the sentence stand alone? If stand alone, please check the wording since the message is not clear. Maybe “We further reveal that membrane or protein scaffold-mediated clustering is crucial since it leads to the unfastening of the inactive NACHT domain conformation.”? Please also check the last sentence of the summary and maybe replace “facilitates”.

A-3.5: We agree that the original sentence could be clearer. We have now rewritten it for better clarity. Additionally, we revised the last sentence in the abstract. We believe this improves the readability of the abstract.

R-3.6. Is activation on anchored NLRP3 still ROS dependent, i.e., sensitive to inhibitors such as PDTC or ebselen?

A-3.6 As suggested by the reviewer, we preincubated primed cells with PDTC and ebselen, and afterwards added triggers nigericin or imiquimod. In Fig. 3f and Supplementary Fig. 13 we show that ROS inhibitors suppress activation of a selected panel of localized NLRP3 variants.

REVIEWERS' COMMENTS

Reviewer #1 (Remarks to the Author):

The authors have addressed all of my previous concerns.

Authors: We thank the reviewer for constructive comments and suggestions that improved the manuscript.

Reviewer #2 (Remarks to the Author):

Boršić et al have addressed all of my major concerns and have produced a very exciting manuscript that demonstrates new interesting findings to the community.

Authors: We thank the reviewer for thoughtful comments that strengthened the manuscript.

I only have a couple of minor image quality/processing concerns that should be addressed:

1) The addition of immunofluorescence to show localisation of NLRP3 following activation improves the manuscript and gives confidence that organelle targeting is maintained. Whilst the data is encouraging, the representative images in the main figure are hard to interpret in places due to quality of the images. If higher quality images are not available, I suggest that separate stains are shown to make this data easier to interpret, similar as seen for example in the supplementary figure 2. If not possible due to space restrictions, the addition of separated images as an additional supplementary figure would be acceptable.

Authors: We thank the reviewer for the suggestion. Live imaging of dying cells is technically challenging, as numerous changes are happening within the cell, resulting in dimming of the organellar stain; in addition, other factors, such as pH alterations, can affect the fluorescence signal. We agree that representative images in the main figures are a bit overcrowded, thus we included separate channel images corresponding to the merged images in the main figures where cells were stimulated with nigericin (Fig. 1c-e; 2c, d) in Supplementary Figures 2 and 5c-d. We believe this improves the visibility and clarity of the data presented. Additionally, we included larger images in the Source Data file.

2) Likewise, supplementary figure 11 shows exciting data suggesting ASC speck positioning is also localised to organelle targeting, but this is hard to see due to the size of the images. Perhaps this figure can be split up to ensure that the images can be seen clearly.

Authors: We thank the reviewer for this observation. To preserve the context and comparability across different conditions, we would prefer to keep these images together within a single supplementary figure rather than splitting them across multiple figures. However, we have included higher-resolution versions of these images in the Source Data file.

3) ASC speck staining in supplementary figure 9 is dim – can the ASC stain be brightened in this figure.

Authors: ASC specks are quite small, which is why we marked them with arrowheads in the images. We have slightly adjusted the brightness and, additionally included larger images in the Source Data file.

Reviewer #3 (Remarks to the Author):

The authors answered all my questions satisfactorily.

Authors: We thank the reviewer for the encouraging comments that helped us improve the study.